

# Field data based validation of an aero-servo-elastic solver for high-fidelity LES of industrial wind turbines

Etienne Muller[1], Simone Gremmo[1], Félix Houtin-Mongrolle[1], Bastien Duboc[2], and Pierre Bénard[1]

[1]Univ Rouen Normandie, INSA Rouen Normandie, CNRS, CORIA UMR6614, 675, avenue de l'Université, Saint-Etienne-du-Rouvray, 76801, Fance
[2]Siemens Gamesa Renewable Energy, 685 Avenue de l'Université, Saint-Etienne-du-Rouvray, 76801, France

**Correspondence:** Etienne Muller (etienne.muller@coria.fr)

**Abstract.** To design the next generations of wind turbines, engineers from the wind energy industry must now have access to new numerical tools, allowing the high-fidelity simulation of complex physical phenomena and thus a further calibration of lower-order models. For instance, the rotors of offshore wind turbines, whose diameter can now exceed 200m, are highly flexible and fluid-structure interactions cannot be neglected any longer. Accordingly, this paper presents a new aero-servo-elastic solver designed to perform high fidelity Large-Eddy Simulations (LES) of wind turbines, as well as of rotor-wake interactions classically occurring in wind farms. In this framework, the turbine blades are modeled as flexible actuator lines. In terms of operating parameters (rotation speed and pitch angles) and power output, the solver is first validated against field data from the Westermost Rough offshore wind farm, for three different operation points. A very good agreement between the numerical results and field data is obtained. To push the validation further, additional results are compared to those given by a certified aero-servo-elastic solver used in the industry, which relies on a Blade Element Momentum (BEM) method. The internal loads throughout the first blade and the deflections at the tip are studied in detail and some discrepancies are observed. Of a reasonable amplitude overall, those are legitimately related to intrinsic modeling differences between the two solvers.

## 1 Introduction

The aerodynamic high-fidelity simulation of an operating horizontal-axis wind turbine is challenging as it features a very high Reynolds number. Therefore one can expect a large range of spatial scales in the flow. Even today, despite the continuous growth of computational power, the direct numerical simulation of such a problem is still out of reach. In contrast, the Large-Eddy Simulation (LES) approach is an interesting compromise to deal with this type of problem, especially when it comes to investigating the complex multi-scale physics of wind-turbine wakes. Although mainly used in academic research, this approach is also attractive for the wind energy industry. Indeed, the results of LES simulations can complement partial field data, when conducting the calibration, validation, and certification of low-order engineering models widely used in the industry. Such numerical results can also provide valuable insights on turbine-wake interactions at the scale of a wind farm. Understanding this phenomenon is indeed critical for siting engineers at the wake losses can impact significantly the annual energy production.



Yet, a LES simulation can remain very costly depending on how the wind turbine is modeled. Using a full 3D model for the rotor, nacelle and tower, necessarily results in constraints on the cell size used in the vicinity of the walls, even more if the boundary layers developing in those regions are to be fully resolved. Regarding authors' knowledge, the latter approach is only reported by Lawson *et al.* in (Lawson et al., 2019). In their work, the wall-resolved simulation of a NREL-5MW wind turbine was performed, both for laminar and turbulent inflows, on a grid counting more than 6 billion nodes. Noteworthy alternative approaches consist in hybrid LES-RANS (Detached-Eddy-Simulation (DES) and derivatives) (Corson et al., 2012; Li et al., 2015; Thé and Yu, 2017) and wall-modeled techniques (Bénard et al., 2018), to achieve a lower time to solution. Still, considering a whole wind farm using one of the aforementioned methods remains computationally unaffordable.

This warrants why the Actuator Line Method (ALM), initially introduced by Shen and Sorensen in 2002 (Sørensen and Shen, 2002), is still a state-of-the-art approach to model a wind turbine at a reasonable cost in a LES framework. Originally, this method was designed to model the rotor blades as simple lines, discretized into 1D elements. The blade element theory is then used to compute the aerodynamic loads applied at the center of each element. The boundary layers around the blades are no longer resolved, thus considerably relaxing the constraints on the local cell size, and avoiding the need to manage rotating or overlapping 3D meshes. To enhance the turbine representation, one can either add the tower and nacelle 3D geometries (Santoni et al., 2017; Ciri et al., 2017; Bénard et al., 2018), or model them as additional actuator lines/disks (Aitken et al., 2014; Churchfield et al., 2015; Gao et al., 2021; Stanly et al., 2022) to limit the extra computational cost.

The simulation of an operating wind turbine becomes even more challenging if one needs to predict the deflections of the blades. As a result, this aspect is still often overlooked in CFD simulations reported in the literature. However, the latest offshore wind turbine prototypes have rotors whose diameter can exceed 200 m. By their composite structure and slenderness, the blades possess a degree of flexibility likely to induce fluid-structure interactions with the surrounding flow. Especially for a flexible blade, turbulent inflows can increase drastically the fluctuations over time of the aerodynamic coefficients, and thus the fatigue (Rezaeiha et al., 2017).

To consider the structural response of a wind turbine to an incident flow, several cost-effective solvers were designed in the wind energy community: OpenFAST (Jonkman et al., 2005; National Renewable Energy Laboratory, 2022), HawC2 (Larsen and Hansen, 2007) and BHawC (Rubak and Petersen, 2005), developed respectively by the National Renewable Energy Laboratory (NREL), the Technical University of Denmark (DTU) and Siemens Gamesa Renewable Energy (SGRE), are typical examples. All these solvers rely on the Blade Element Momentum (BEM) method to compute the aerodynamic forces. Although predictive, these tools are limited by their inability to simulate the surrounding flow and especially the wakes. As such, one can no longer qualify these solvers as true CFD tools. To address the various Design Load Cases (DLC) referenced in the standards from the International Electrotechnical Commission (IEC), an appropriate modeling of the incoming flow, capturing the main physical mechanisms involved, is needed. For instance, assessing the loads experienced by a turbine operating in the wake of an upstream one is of critical importance. In such a scenario, the Dynamic Wake Meandering (DWM) model, initially introduced by Larsen *et al.* (Larsen et al., 2007), can be used to mimic the dynamic of an incoming wake. Because they also feature a very low time to solution, these tools and wind models altogether are extensively used by wind-turbine manufacturers



to drive the design iterations or siting studies. Yet, careful calibrations of all models are needed for them to be certified and thus usable. This can be achieved by validating the predictions with field data and/or results coming from high-fidelity tools.

The latter, significantly more computationally intensive, mainly consist in a two-way coupling between a vortex-based or a CFD-like aerodynamic solver and a structural solver. The first one is in charge of getting the flow field, while the second one computes the structural response of the turbine. The literature already counts numerous works where this kind of approach is described (Lee et al., 2012; Li et al., 2015; Meng et al., 2018; Sprague et al., 2020; Hodgson et al., 2021; Elie et al., 2022; Della Posta et al., 2022).

In this paper, we present the implementation, verification and validation of a new solver derived from such a coupling: the high-order LES code YALES2 (Moureau et al., 2011) is used to solve the flow and compute the loads acting on the blades using the ALM, and the structural response is computed by BHawC. A substantial advantage of this code is the possibility to emulate the actual controller of industrial wind turbines. To the authors' knowledge, (Gremmo et al., 2022) is the only reference where a code with these capabilities has been reported. In this study, the authors investigated a row of seven industrial wind turbines, operating in several wind conditions.

The paper is organized as follows. First of all, the two codes involved in the coupling are briefly described in Section 2, after which Section 3 presents thoroughly the coupling implementation. The verification of the kinematic and dynamic aspects involved in the coupling is then reported in Section 4. Finally, Section 5 presents the solver validation based on the simulation of an isolated turbine of the Westermost Rough offshore wind farm. Several operation points are considered in terms of wind speed and turbulence intensity. For each of them both YALES2 and YALES2-BHawC are used to model the wind turbine. The numerical predictions of the turbine overall performance are compared with field data. Additional numerical results, representative of the structural response, are also analyzed and compared in more depth.

## 2 Solvers involved in the coupling

### 2.1 YALES2: an aerodynamic solver

#### 2.1.1 Governing Equations

In the context of wind energy related LES simulations, one can solve the filtered Navier-Stokes equations for incompressible flows. Noting $\tilde{\bullet}$ the implicit spatial filtering resulting from the mesh resolution, these equations can be formally written as:

$$\boldsymbol{\nabla} \cdot \tilde{\boldsymbol{u}} = 0, \tag{1}$$

$$\frac{\partial \tilde{\boldsymbol{u}}}{\partial t} + (\tilde{\boldsymbol{u}} \cdot \nabla) \tilde{\boldsymbol{u}} = -\nabla \tilde{P} + \nu \nabla^2 \tilde{\boldsymbol{u}} + \nabla \cdot \underline{\tau}_{\mathrm{SGS}} + \boldsymbol{f}, \tag{2}$$

where $\nu$ is the kinematic viscosity, $\tilde{P}$ the reduced pressure, and $\boldsymbol{f}$ a body force such as the gravity for instance. The stress tensor $\underline{\tau}_{\mathrm{SGS}}$ results from the filtering operation and is a function of the unsolved sub-grid scales (SGS) in the flow. In the present work, this tensor is assessed using the localized version of the dynamic model of Smagorinsky (Lilly, 1992).



### 2.1.2 Solver description

YALES2 (Moureau et al., 2011) is an in-house massively-parallel finite volume library intended for solving various fluid
mechanics related problems. YALES2 includes, among others, an incompressible solver for carrying out LES simulations of
wind turbine wakes (Bénard et al., 2018).

When considering the equations given in section 2.1.1, the pressure-velocity coupling is handled by a projection method (Chorin,
1968). In this framework, the Poisson equation for the pressure is solved with the Deflated Preconditioned Conjugate Gradient
method reported in (Malandain et al., 2013). Additionally, the solver makes use of a 4th-order 2-step Runge-Kutta method to
carry out the time integration. Likewise, the central scheme used for spatial discretization is 4th-order. To enhance the com-
puting performance, an in-house two-level grid partitioning is used (Moureau et al., 2011). The first level consists in splitting
the computational domain into *partitions*. A partition is a fraction of the computational domain, which is fully managed by
one MPI rank. Each partition is then further decomposed into a collection of *cell groups*. The number of cells included in each
group is set to avoid any cache-memory miss.

### 2.1.3 Wind turbine modeling

In YALES2, the ALM is used to model operating wind turbines. As stated in the introduction, the approach allows to model
the forces applied by wind turbine blades on an incident flow. It is particularly suitable for investigating the physics of wind
turbine wakes at a reasonable cost. Each blade geometry is replaced by a simple line, discretized into 1D elements of equal
width $w$. The center point of an element is called *particle*. The particle location is also chosen to match the quarter of chord of
the corresponding airfoil. In order to ease its handling, the rotor simplified geometry relies on a set of 3D bases:

- The *rotor basis* (RB): linked to the rotor center, this basis allows to represent the tilt and yaw angles. However, its
  orientation remains independent of the azimuthal rotation.

- The *blade bases* (BB): their origins coincide with the blade roots. They translate accurately the orientation of each blade,
  by taking into account their azimuthal position and the pitch angle. The turbine coning can also be represented by their
  means.

- The *particle bases* (PB): they are attached to each blade element. They allow to represent the local twist angle, as well
  as the prebend and sweep of the blades.

All these additional bases, illustrated in Figure 1, are given in the global (GB) reference frame of YALES2 (Y2). For instance,
the rotor basis can thus be written as the transition matrix $\mathrm{T}_{GB\,Y2}^{RB\,Y2}$. This natural framework proves very advantageous for
navigating from one basis to another. Initially, the particle bases are linearly interpolated in a lookup table, which define the
blade undeformed geometry and contains similar bases associated with specific spanwise positions along the blade.





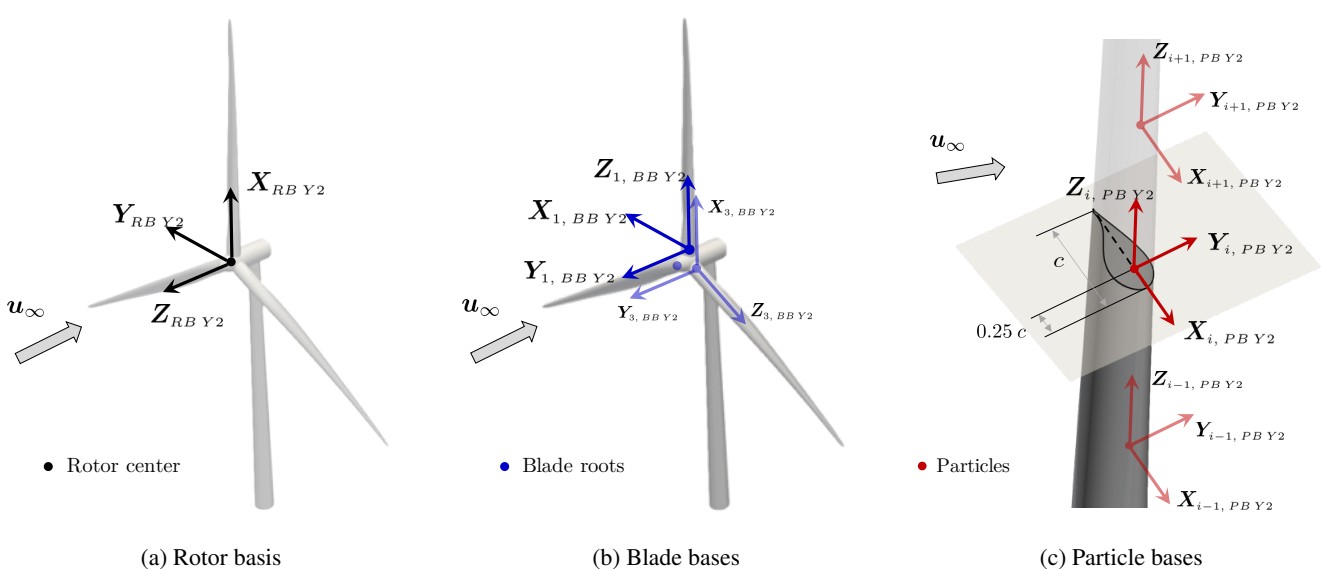

| (a) Rotor basis | (b) Blade bases | (c) Particle bases |
| --- | --- | --- |

**Figure 1.** Additional set of bases used in the ALM framework of YALES2.

The lift $L$ and drag $D$ forces, assumed constant on each blade element, are computed at the particle location as follows:

$$L = \frac{1}{2}\,\rho\,u_{rel}^2\,c(s)\,w\,C_L(\alpha,s), \tag{3}$$

$$D = \frac{1}{2}\,\rho\,u_{rel}^2\,c(s)\,w\,C_D(\alpha,s), \tag{4}$$

120 where $u_{rel}$ is the magnitude of the fluid velocity seen by the moving airfoil and $\rho$ is the fluid density. The lift and drag coefficients, respectively denoted $C_L$ and $C_D$ here, are interpolated linearly in the aforementioned lookup table, which provides them as a function of both the angle of attack $\alpha$ and the spanwise position $s$ along the blade. The airfoil chord $c$ is also retrieved from the same lookup table. The velocity $u_{rel}$ is further defined as a combination of both the local fluid velocity $\boldsymbol{u}_f$ and the particle velocity $\dot{\boldsymbol{x}}_P$:

125 $$\boldsymbol{u}_{rel} = \boldsymbol{u}_f - \dot{\boldsymbol{x}}_P. \tag{5}$$

Once given in the particle frame, this velocity enables to compute the angle of attack $\alpha$.

Yet, the forces obtained this way are singular and must be smoothed before being added as a body force in the Navier-Stokes equations. In the ALM framework, this operation is known as the mollification process. It acts as a spatial filtering operation, typically based on a Gaussian-like kernel $\eta$ (Sørensen and Shen, 2002):

130 $$\eta(d) = \frac{1}{\epsilon^3 \pi^{3/2}} \exp\left[-\left(\frac{d}{\epsilon}\right)^2\right], \tag{6}$$

where $d$ is the distance to the kernel center, and $\epsilon$ is the kernel radius. The latter is usually chosen as twice the maximum cell-size encountered in the rotor region.



For an exhaustive description of the ALM implementation in YALES2, the reader is referred to the PhD thesis of Houtin-Mongrolle (Houtin-Mongrolle, 2022). Issues regarding the optimization of the ALM computational cost are also covered.

## 2.2 BHAWC: an aero-servo-elastic solver

BHawC (Rubak and Petersen, 2005) is a nonlinear aero-servo-elastic solver intended to support the design and certification of wind turbines. Validated continuously against field data, the code allows a fast assessment of a wind turbine structural response to external loads, so as to investigate numerous Design Load Cases (DLC) in a reasonable time. Indeed, for an appropriate discretization of the structure, the code can provide a 1:1 time to solution[1]. Yet, the code can only handle one turbine at a time as it considers only one incident flow field. The simulation of an operating wind farm remains feasible by considering different incident flows for each turbine, mimicking at best the local flow properties.

The modeling of a wind turbine consists of several substructures used to represent the foundation, tower, nacelle, hub, drivetrain, shafts, as well as the rotor blades. Most of the substructures are modeled with equilibrium-based non-homogeneous anisotropic beam elements counting 12 degrees of freedom: 3 rotations and 3 translations per node (Krenk and Couturier, 2017). The drivetrain, as well as the different bearings, are modeled as purely torsional elements. Similarly to YALES2, BHawC also makes use of additional frames to manage the mentioned sub-structures and the elements used to discretize them. In particular, all the bases discussed in Section 2.1.3 have their twin in BHawC (BH), even though their orientation can differ. This is the case for the blade bases, depicted in Figure 2.

All the structural degrees of freedom, as well as their velocity and acceleration, are solved in BHawC (BH) global frame, which is attached to the bottom of the turbine foundation. The vector containing all the degrees of freedom is denoted by $\boldsymbol{x}(t)$, and its first and second derivative by $\dot{\boldsymbol{x}}$ and $\ddot{\boldsymbol{x}}$ respectively. BHawC targets the following equilibrium at each time-step (Skjoldan, 2011):

$$\boldsymbol{f}_{\text{iner}}(\boldsymbol{x},\dot{\boldsymbol{x}},\ddot{\boldsymbol{x}}) + \boldsymbol{f}_{\text{damp}}(\boldsymbol{x},\dot{\boldsymbol{x}}) + \boldsymbol{f}_{\text{int}}(\boldsymbol{x}) = \boldsymbol{f}_{\text{ext}}(\boldsymbol{x},\dot{\boldsymbol{x}},\ddot{\boldsymbol{x}}), \tag{7}$$

where $\boldsymbol{f}_{\text{iner}}$, $\boldsymbol{f}_{\text{damp}}$, $\boldsymbol{f}_{\text{int}}$, and $\boldsymbol{f}_{\text{ext}}$ are the inertial, viscous damping, internal, and external loads respectively. Especially, the latter encompasses the aerodynamic loads. To solve the previous equation, the Newton-Raphson method is used to derive its residual form and iterate over it:

$$\underline{\boldsymbol{M}}(\boldsymbol{x})\delta\ddot{\boldsymbol{x}} + \underline{\boldsymbol{C}}(\boldsymbol{x},\dot{\boldsymbol{x}})\delta\dot{\boldsymbol{x}} + \underline{\boldsymbol{K}}(\boldsymbol{x},\dot{\boldsymbol{x}},\ddot{\boldsymbol{x}})\delta\boldsymbol{x} = \boldsymbol{r} = \boldsymbol{f}_{\text{ext}} - \boldsymbol{f}_{\text{iner}} - \boldsymbol{f}_{\text{damp}} - \boldsymbol{f}_{\text{int}}, \tag{8}$$

where $\underline{\boldsymbol{M}}$, $\underline{\boldsymbol{C}}$, and $\underline{\boldsymbol{K}}$ are the mass, damping, and stiffness matrices respectively. Additionally, the Newmark method is used to advance the system in time and relate $\ddot{\boldsymbol{x}}$ and $\dot{\boldsymbol{x}}$ to $\boldsymbol{x}$. This solution procedure leads to a linear system at each Newton iteration, where the correction vector $\delta\boldsymbol{x}$ is the only unknown. A LU factorization method is typically used to invert the system. Once $\delta\boldsymbol{x}$ is factored in $\boldsymbol{x}$, the code updates the aforementioned matrices and all the components of the residual $\boldsymbol{r}$ except for the aerodynamic loads.

---

[1]Simulating one second of physical time requires one second of wall clock time.



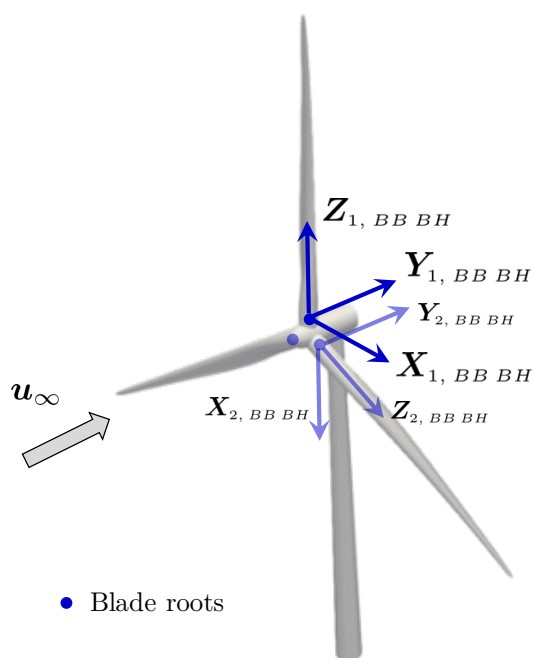

**Figure 2.** Blade bases in BHawC.

Those are computed only once per time step, prior to the first Newton iteration. To do so, BhawC relies on a *Blade Element Momentum* (BEM) method (Sørensen, 2016), enhanced by many corrections. For instance, BHawC benefits from a modeling
of the dynamic stall phenomenon (Leishman and Beddoes, 1989) and from the Prandtl correction for the tip/root losses, to cite only a few.

Finally, a complete framework is available in BHawC to include an emulation of a real controller in the wind turbine modeling. As a result, the rotor rotation speed is intrinsically unsteady. This approach also allows to update other operating parameters in time, especially the pitch angles.

**3  YALES2-BHawC: a coupled high-fidelity aero-servo-elastic solver**

As implemented in YALES2, the ALM models the blades of a wind turbine as fully rigid lines, animated only by rotational movements. The azimuthal rotation speed is usually imposed as constant. This ALM implementation allows to simulate the flow surrounding an arbitrary number of wind turbines. Phenomena such as wake steering, wake meandering, and wakes combination can therefore be studied with YALES2. However, the structural response of a wind turbine to an incident flow
remains unknown. Conversely, BHawC can provide the structural response of an industrial wind turbine, for a wide variety of incident flows, while also emulating real control strategies, meaning the azimuthal rotation speed is unsteady. Yet, the code does not offer any information on the resulting wake, and only one turbine can be handled at a time. In light of this, a coupling



between YALES2 and BHawC was thus interesting to get accurate insights on the flow surrounding a turbine and the fluid-solid interactions involved when this turbine operates.

As designed, the coupling between YALES2 and BHawC only affects the blades of the wind turbine. The remaining sub-structures, such as the nacelle and the tower, are still represented in BHawC, but the aerodynamic loads they would normally experience are disabled. In the following, the coupling strategy is described in details, as well as the mathematical framework upon which the coupling relies to transfer data between YALES2 and BHawC. Additional technical informations regarding the coupling itself can be found in Appendix A.

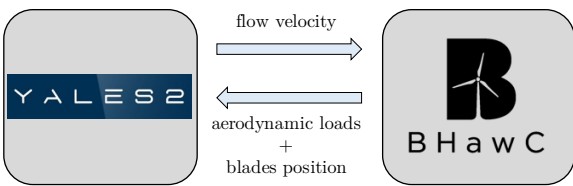

(a) Strategy 1: the BEM method implemented in BHawC remains necessary.

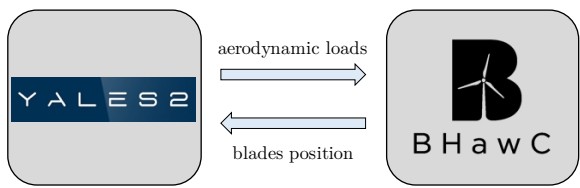

(b) Strategy 2: YALES2 is a total substitute to the BEM method.

**Figure 3.** Possible strategies to adopt for the coupling between YALES2 and BHawC. The second one was finally retained because it is more generic.

Regarding the coupling strategy, two options were available, which are depicted in Figure 3. The first one (cf. Figure 3a) is for instance employed in (Lee et al., 2012; Sprague et al., 2020; Hodgson et al., 2021; Elie et al., 2022). In this strategy, the structural solver (BHawC) first provides the position of the blade elements to the aerodynamic solver (YALES2), which can then assess the flow velocity at these positions. The flow velocity is returned to the structural solver, and then combined with the blade elements velocity to derive the relative velocity $\boldsymbol{u}_{rel}$. The blade element theory, which is part of the BEM method,

is then used to compute the aerodynamic loads acting on the blades. The loads contribute to the movement and deflections of the blades in the structural solver, but they are also sent back to the aerodynamic solver to be mollified on the Eulerian grid. From the authors' perspective, this approach is however not generic enough for several reasons. First, it cannot work without the structural solver also having at least the blade element theory implemented within. In other words, a standalone structural solver is not suited for this strategy. Second, assuming that it is desired to also couple the tower and nacelle of the turbine, this

strategy requires the structural solver to compute the loads acting on these sub-parts in the same way as for the blades, meaning by use of the blade element theory. Third, if the tower and nacelle were to be body fitted in the aerodynamic solver, deriving



the appropriate fluid velocities to send to the structural solver would not be straightforward anymore, as blockage effects would be fully represented in the vicinity of these sub-parts.

Given these limitations, the second coupling strategy was selected (cf. Figure 3b). In essence, BHawC communicates the
current position and velocity of the blade elements to YALES2, and the ALM framework is used to compute and mollify the aerodynamic loads. The loads are also returned to BHawC to increment the position and the deflections of the blades. One should note that this second approach completely bypasses the BEM method implemented in BHawC. This type of strategy is also used in (Li et al., 2015; Meng et al., 2018). In detail, the computation of the aerodynamic loads requires more information. In particular, the transition matrix $\mathrm{T}_{GB\,Y2}^{PB\,Y2}$ must be known at each time-step and for each blade element. This matrix represents
the element bases, as given in YALES2 global frame. It can be obtained as a chain of transition matrices:

$$\mathrm{T}_{GB\,Y2}^{PB\,Y2} = \underbrace{\mathrm{T}_{GB\,Y2}^{GB\,BH}}_{(1)} \cdot \underbrace{\mathrm{T}_{GB\,BH}^{BB\,BH}}_{(2)} \cdot \underbrace{\mathrm{T}_{BB\,BH}^{PB\,BH}}_{(3)} \cdot \underbrace{\mathrm{T}_{PB\,BH}^{PB\,Y2}}_{(4)} \cdot \tag{9}$$

The matrix (1) is computed only once as:

$$\mathrm{T}_{GB\,Y2}^{GB\,BH} = \mathrm{T}_{GB\,Y2}^{RB\,Y2} \cdot \mathrm{T}_{RB\,Y2}^{GB\,BH} \cdot \tag{10}$$

The rotor deformations and rigid movements are managed directly in BHawC. This makes the rotor basis in YALES2 com-
pletely steady during the simulation. Thus, the second term on the right hand side can be hard coded. The matrices (2) and (3) are computed and sent by BHawC at every time-step. Despite the particle bases in YALES2 (PB Y2) sharing the same overall orientation than those of the aerodynamic nodes in BHawC (PB BH), the matrix (4) differs from the identity matrix. In a coupled simulation, the number of blade elements is enforced by YALES2 in order to guarantee their even distribution along the blade. This follows from the chosen mollification kernel (cf. eq. (6)), whose radius does not depend on the spanwise
position. In this context, BHawC is unaware of the twist angle evolution along the blades, and assumes it to be uniformly zero. In YALES2, the element bases remain set up as in a standalone simulation: their orientation therefore includes the twist angle. As the modeled airfoils are assumed fully rigid, this matrix is then computed only once as follows:

$$\mathrm{T}_{PB\,BH}^{PB\,Y2} = \underbrace{\left[\mathrm{T}_{BB\,BH}^{PB\,BH}\right]_{t=0}^{-1}}_{(i)} \cdot \underbrace{\mathrm{T}_{BB\,BH}^{BB\,Y2}}_{(ii)} \cdot \underbrace{\mathrm{T}_{BB\,Y2}^{PB\,Y2}\big|_{t=0}}_{(iii)} \cdot \tag{11}$$

As the matrix (3), the matrix (i) is computed and sent by BHawC. The matrix (ii) is trivially derived from the figure Figures 1b
and 2, while the matrix (iii) is interpolated from a lookup table (cf. Section 2.1.2).

The computation of the aerodynamic loads in YALES2 also requires to get the relative flow velocity at the particles position. As shown in Figure 4, the particles are placed on a first axis, here called the *aerodynamic axis* (AA) for the sake of clarity. This axis differs from the *elastic axis* (EA) defined in BHawC, on which the corresponding aerodynamic nodes are located. Specifically, these nodes are positioned on the structural elements, which define the discretized elastic axis. During a coupled
simulation, BHawC communicates to YALES2 the velocity and position of the aerodynamic nodes only. In that sense, the vector $\overrightarrow{NP}$ is needed for each blade element to: (1) interpolate the fluid velocity at the particle actual position, (2) deduce the



particle velocity from the one of the aerodynamic node:

$$\dot{\boldsymbol{x}}_{P,GBY2} = \dot{\boldsymbol{x}}_{N,GBY2} + \boldsymbol{\Omega}_{GBY2} \times \left( \mathrm{T}_{GBY2}^{PBY2} \cdot \overrightarrow{NP}_{PBY2} \right), \tag{12}$$

where $\boldsymbol{\Omega}_{GBY2}$ is the rotation vector of the airfoil, given in YALES2 global frame.

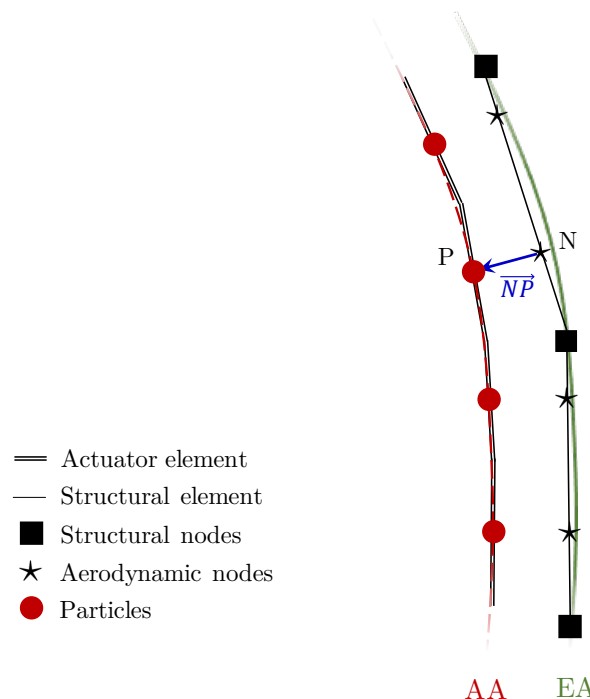

**Figure 4.** Sketch of the blade discretization, illustrating the distinction between the aerodynamic axis (AA) and the elastic axis (EA).

As with the transition matrix $\mathrm{T}_{PB\,BH}^{PB\,Y2}$, $\overrightarrow{NP}$ can be computed only once in the corresponding element basis, where it should remain constant afterwards. Once computed, the aerodynamic forces $\boldsymbol{F}$ and moments $\boldsymbol{M}$ are transferred back to the aerodynamic nodes and returned to BHawC. Here again, the vector $\overrightarrow{NP}$ is used to transport the moment from the AA to the EA :

$$\boldsymbol{M}_{N,PBY2} = \boldsymbol{M}_{P,PBY2} + \overrightarrow{NP}_{PBY2} \times \boldsymbol{F}. \tag{13}$$

At the beginning of each of its time-steps, BHawC predicts the velocities and positions of the blade elements. The converged velocities and accelerations from the previous time-step are used for this purpose. In addition, the next orientation of the element bases is also predicted. These data are then sent to YALES2. However the corresponding variables are not updated immediately, so as to evaluate a relative velocity which accounts for the last mollification of the aerodynamic forces. Angles of attack are inferred using the element bases from the previous time-step, and the aerodynamic loads are derived subsequently in these bases. The position, velocity and basis of each element is then updated to transfer the loads into both the global frame of



YALES2 and BHawC:

$$\boldsymbol{F}_{N,GB\,BH} = \left[\mathrm{T}_{GB\,Y2}^{GB\,BH}\right]^{-1} \cdot \mathrm{T}_{GB\,Y2}^{PB\,Y2} \cdot \boldsymbol{F}_{N,PB\,Y2},$$
$$\boldsymbol{M}_{N,GB\,BH} = \left[\mathrm{T}_{GB\,Y2}^{GB\,BH}\right]^{-1} \cdot \mathrm{T}_{GB\,Y2}^{PB\,Y2} \cdot \boldsymbol{M}_{N,PB\,Y2}. \tag{14}$$

The forces $\boldsymbol{F}_{N,GB\,BH}$ and moments $\boldsymbol{M}_{N,GB\,BH}$ thus obtained are sent back to BHawC, and the structural solver iterates to
find the next position and shape of each blade. The aerodynamic loads are kept constant in all Newton iterations to significantly speed up the simulation. This approximation is motivated by the small azimuthal angle swept by the blades during a time-step. Meanwhile, the mollification of the forces takes place in YALES2, at the *predicted* position of the blade elements. The one or two iterations usually needed for the structural solver to converge warrants this second approximation. As previously described, the advancement of a blade is further illustrated in Figure 5.

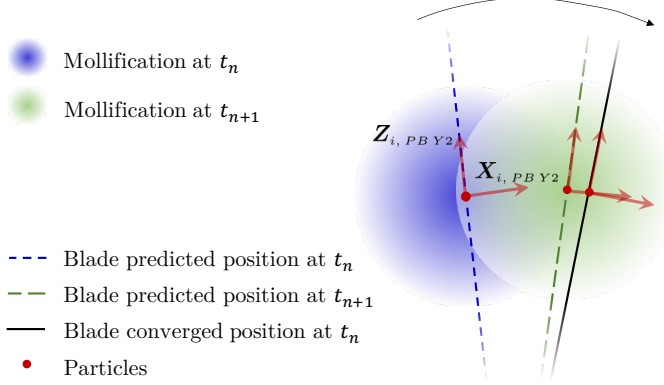

**Figure 5.** Displacement of a blade portion during a time-step. The predicted and final positions were made significantly different only for the sake of clarity, but in practice they almost overlap.

For an adequate trade-off between accuracy, stability, and time to solution, the time-step size commonly used in BHawC is $\Delta t_{BH} = 0.02$ s. However, applying the CFL conditions (related to both the local fluid velocity and blade-element velocities) in YALES2 can provide significantly larger time steps, depending on the far field wind speed and mesh used. Therefore, imposing $\Delta t_{Y2} = \Delta t_{BH}$ leads to a computational overhead in most cases. This motivated the development of a sub-stepping mechanism to circumvent this situation. Specifically, the time-step in YALES2 is allowed to be a multiple of the one used in BHawC,
within the limit of $\Delta t_{Y2} = 3\,\Delta t_{BH}$ for stability issues. In this context, the aerodynamic loads are computed only once, during the first sub-step. During the subsequent one(s), YALES2 uses the predicted element bases to update the loads orientation, before sending them to BHawC. While BHawC corrects the blades shape and position, YALES2 performs the mollification of the aerodynamic forces at the predicted location of the blade elements.

YALES2-BHawC is thus a partitioned loosely-coupled solver. The coupling implementation is a combination of the Conven-
tional Serial Staggered (CSS) and Conventional Parallel Staggered (CPS) procedures described by Farhat and Lesoinne (Farhat and Lesoinne, 2000) (see Figure 6). The CPS procedure aims at reducing the total computational cost of the coupled simulation





by allowing the concurrent execution of the aerodynamic and structural solvers. However, its stability and accuracy require a sufficiently small time step (Farhat and Lesoinne, 2000). In the coupled simulations carried out, the structural solver occasionally diverged for wind speeds in the range from 8 to 12 m/s at hub height. We expect this issue to be at least partially removed

by implementing an appropriate dynamic stall model in YALES2. Indeed, simulations carried out with the standalone version of BHawC showed a link between the divergence of the structural solver and the dynamic stall phenomenon model activation. It is emphasized that the induction is close to being maximum in the mentioned wind speed range, leading to a strong coupling between the structure and the ambient flow. Therefore, the relaxation of aerodynamic forces induced by the dynamic stall phenomenon is likely to be essential in these cases.

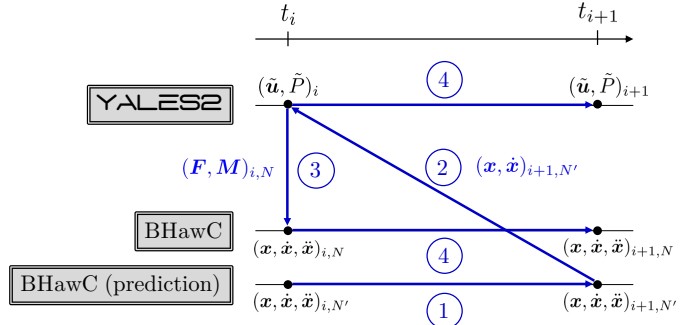

**Figure 6.** Coupling procedure used between YALES2 and BHawC.

**4   Verification**

Before attempting real-application cases, the coupled code YALES2-BHawC has been extensively checked. Relying on appropriate unit tests, we verified both the kinematic and dynamics aspects involved in the coupling.

Discussed in Section 4.1, the kinematic verification is carried out by comparing the results from simulations run with YALES2 standalone (Y2) and YALES2-BHawC (Y2-BH) respectively. On the other hand, the verification of the dynamic

aspects, reported in Section 4.2, is supported by a comparison of the results obtained with BHawC standalone (BH) and YALES2-BHawC. In all cases we consider the SWT-6.0-154 wind turbine model (Siemens Gamesa Renewable Energy, 2014), whose rotor diameter is $D = 154$ m.

The rotor rotation speed is fixed to a constant value in YALES2 standalone, as it is the most straightforward approach. However, this cannot be enforced as easily in BHawC and in the coupled code. Indeed, BHawC needs a controller component

to adjust the rotation speed. A simulation can still be run without, but the rotation speed will then increase progressively without any limitation. Given the structural modifications of the blades that were also necessary to complete the verification activities, targeting a given rotation speed in BHawC with the actual controller was considered too challenging. We thus replaced this controller by a *dummy* one, designed to track a user-supplied rotation speed by adjusting the rotor torque only.



In both YALES2 and BHawC, the incident flow is chosen uniform with no synthetic turbulence added, and its speed is set to
8 m/s. Besides, for the sake of similarity, we nullified the induction factors in all simulations. In YALES2, the resulting flow is
thus uniform throughout the whole domain, which allows the use of a narrow computational domain (as illustrated in Figure 7),
along with a very coarse cartesian grid (cell size $\sim 12.5$ m).

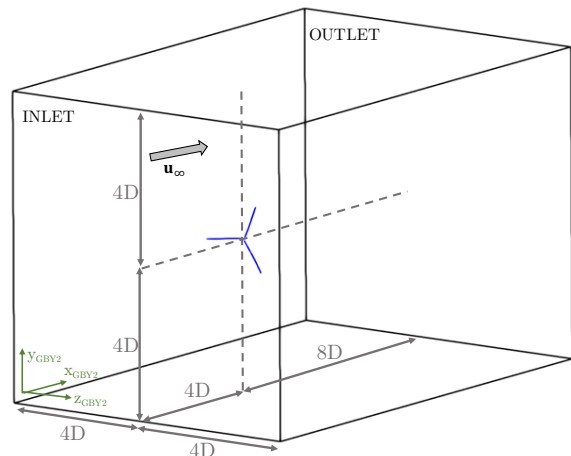

**Figure 7.** Geometry of the computational domain used with YALES2 and YALES2-BHawC during the verification campaign.

In all numerical setups, the blades are equally discretized: 75 aerodynamic nodes/particles are evenly distributed along the
blades span. The look-up table used in YALES2 was built to contain the blade data BHawC would attach to its aerodynamic
nodes in such a configuration. This way, blade data at the particles location in YALES2 are enforced to match those used in
BHawC, as interpolation errors are avoided.

Besides, the blades elasticity and shear moduli are chosen 1000 times larger than the actual ones, almost nullifying the
blade deflections. Obviously, this does not apply to YALES2 standalone simulations where blades are fully rigid. Consistently,
the remaining wind-turbine sub-structures present in the structural solver are also modeled as *almost* fully rigid, and the
aerodynamic loads acting on them are nullified.

## 4.1   Kinematic aspects

As explained in Section 3, in a coupled simulation BHawC communicates to YALES2 the position and velocity of all AL
particles, as well as the 3D basis attached to them. Therefore, we need to ensure, at first, that the AL particles in YALES2-
BHawC are moving as they would in YALES2 standalone. The results obtained with both codes on the configuration previously
described are thus compared. However, two differences between the numerical setups must be stressed, as they cannot be
removed. The first one concerns the blades: they are *fully* rigid in YALES2, while they *almost* are in YALES2-BHawC. As
a result, the blades in the coupled simulation necessarily experience tiny deflections. The second difference is related to the
rotor rotation speed: it is imposed constant throughout the whole simulation in YALES2, while a transient state occurred in

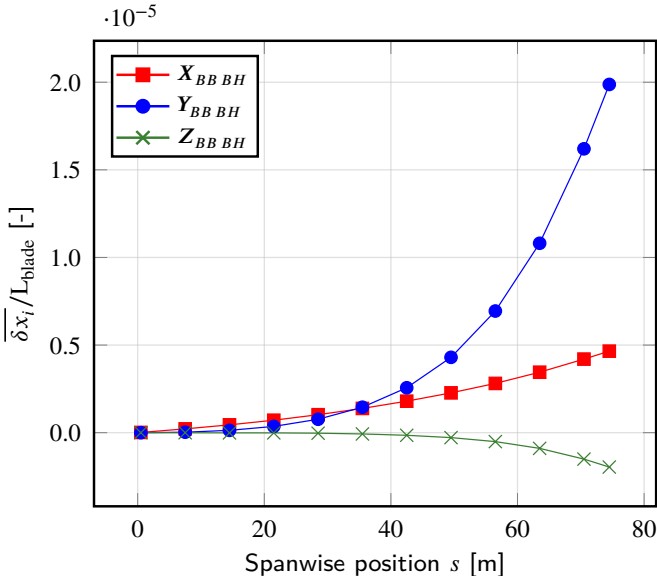

**Figure 8.** Nondimensional mean error on the particles position $x_P$ in BHawC blade basis, given as a function of the spanwise position. The corresponding standard deviations are represented as error bars (not visible).

the coupled code because of the dummy controller. For this reason, we waited for the rotation speed to converge in YALES2-
BHawC before considering the results. In the following, we display the mean error and corresponding standard deviations obtained between the two codes for various quantities of interest. A mean error is defined as follows:

$$\overline{\delta \bullet} = \frac{1}{\theta_N} \int_0^{\theta_N} |\bullet_{\text{Y2-BH}} - \bullet_{\text{Y2}}| \, d\theta \tag{15}$$

where $\bullet$ is one of the quantity of interest, and $\theta_N$ is the total azimuth angle swept by the blade during the $N$ last full revolutions of the rotor. For this verification study, N was set to 10.

First, the effect of the residual blade deflection, on the particle position, should be quantified. Figure 8 shows the average difference on the particles $x$, $y$, $z$ coordinates in the blade basis nondimensionalized by the blade length $L_{\text{blade}} = 75$ m. The maximum deformation occurs at the tip of the blade and is oriented in the flapwise direction $\boldsymbol{Y}_{BB\,BH}$, which is also the streamwise direction. As expected, this deformation is still very small compared to the length of the blade. The spanwise stretching (direction $\boldsymbol{Z}_{BB\,BH}$) and edgewise bending (direction $\boldsymbol{X}_{BB\,BH}$) are about ten times smaller, and thus neglected in
the following analysis. Standard deviations are also given in the figure, but barely (if not) visible as they are extremely small.

Similarly, the orientation of the particle bases is also checked. To perform the comparison, we compute the dot product between the respective particle-basis $x$, $y$, $z$ vectors of the YALES2 standalone and coupled simulations. As depicted in Figure 9, the dot products are all very close to 1, which means that the particle bases are equally oriented. The standard deviations are also provided here, but again too small to display. These first results lead to conclude that the particles position
and related bases are correctly retrieved from the data sent by BHawC in a coupled simulation.

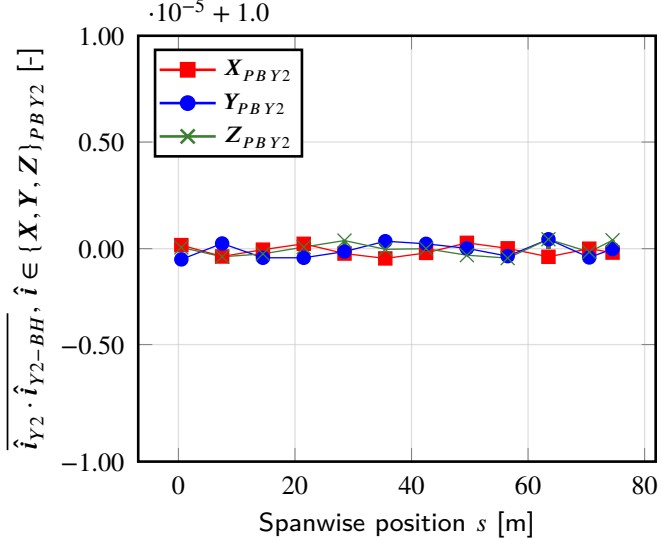

**Figure 9.** Mean dot products between the particle bases obtained in YALES2 and YALES2-BHawC. Results are given as a function of the spanwise position. The associated standard deviations are represented as error bars (not visible).

The next step is to check the particles velocity, as they are also part of the communications between BHawC and YALES2 (cf. eq. (12)). Figure 10 shows the errors for the three velocity components, expressed in the global basis of YALES2. A particle velocity in the streamwise direction (direction $X_{GBY2}$) is due to the tilt angle and to the potential flapwise deflection of the blades. The discrepancies observed on this velocity component are about 10 times smaller than those for the other two components that mainly evolve in the rotor plane. For these two, the error grows linearly with the blade span to be maximum at the tip. This is easily related to the remaining flexibility of the blade in YALES2-BHawC. Overall, the mean errors and standard deviations remain small compared to the modulus of the rotational speed at the tip $u_{\text{tip}} \approx 75$ m/s).

Finally, the angle of attack computed at each particle location is also looked at, as it contributes to the aerodynamic loads computation. The discrepancies obtained between the two approaches are presented in Figure 11. It can be observed that they remained below $0.012°$ on all the blade span, which is completely acceptable.

## 4.2 Dynamic aspects

In order to consider the coupling implementation bug-free, an extra analysis of the aerodynamic loads is required. Indeed, the dummy controller used with YALES2-BHawC could make the rotation speed reach its target even though the predicted aerodynamic torque was completely wrong. Likewise, rigid blades cannot reflect a possibly poorly predicted thrust since their flapwise deflection would be close to zero.

To carry on this analysis, the internal loads computed by BHawC and YALES2-BHawC are compared. Indeed, with the induction removed in both cases, the BEM and AL methods should compute identical aerodynamic loads, as they are both

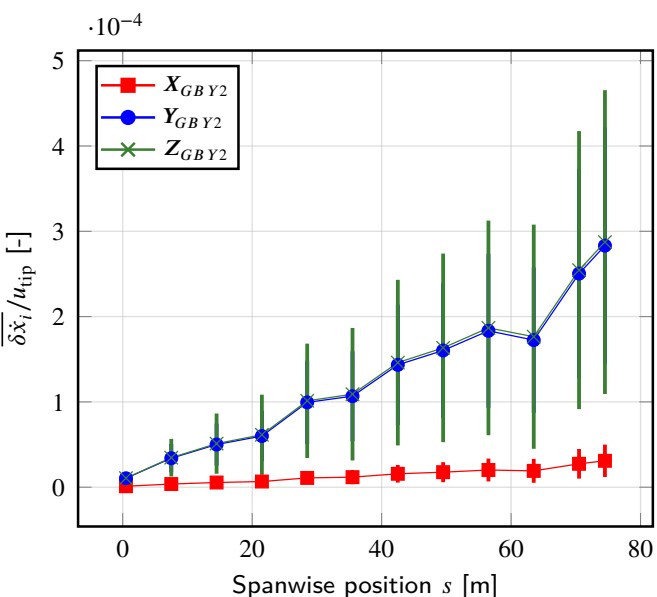

**Figure 10.** Nondimensional mean errors on the particles velocity $\dot{x}_P$, given in YALES2 global basis. Results are given as a function of the spanwise position. The corresponding standard deviations are represented as error bars.

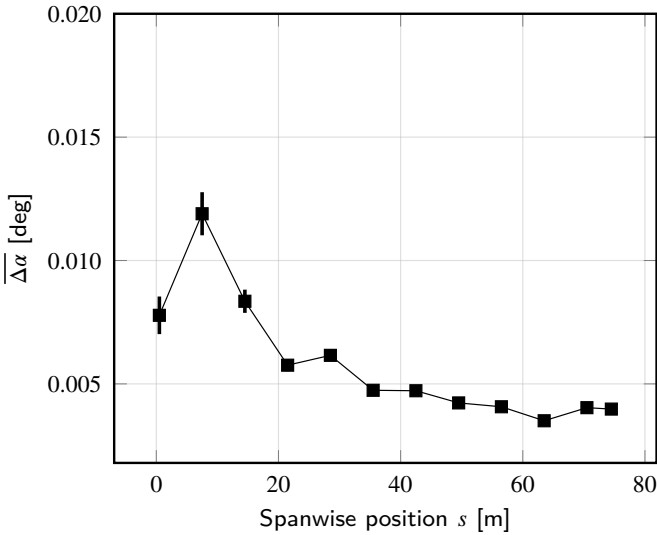

**Figure 11.** Mean error on the angle of attack $\alpha$, as a function of the spanwise position. The corresponding standard deviations are represented as error bars.





downgraded to a simple polar reading. Consequently, the resulting internal loads computed by each code are also expected to be exactly the same. Here again, we use the configuration described at the beginning of the current section.

In both approaches, we consider the internal forces $F_i$ and moments $M_i$ (with $i \in \{x, y, z\}_{BB\,BH}$) in blade sections located at different spanwise positions $s$, to build the following mean relative error:

$$\overline{\Delta \bullet}_\theta = \frac{1}{\theta_N} \int_0^{\theta_N} \left| \frac{\bullet_{\text{Y2}-\text{BH}} - \bullet_{\text{BH}}}{\bullet_{\text{BH}}} \right| \mathrm{d}\theta, \tag{16}$$

where $\bullet$ stands for either $F_i$ or $M_i$, and $\theta_N$ is defined as in Section 4.1.

The results are given in Figure 12. Overall the relative errors are very low, with values less than $0.001\%$. The same observa-
tion applies to the corresponding standard deviations.

Explaining the remaining errors is not straightforward. In contrast to Section 4.1, the small deformations of the structure cannot be blamed as they exist in both simulations. However, the linear system solved by the structural solver proved to be ill-conditioned, likely because of the artificial stiffness added to the structure. Therefore, the structural responses could still significantly deviate even though the initial discrepancies in the aerodynamic loads were close to the machine single precision.
Such differences are expected as BHawC and YALES2-BHawC rely on input files that are formatted differently. Nevertheless, the relative errors presented are deemed small enough to consider the coupled code fully operational.

## 5    Validation

This section is centered on the validation of the coupled code YALES2-BHawC. To carry it through, we consider the case of an isolated turbine of the Westermost Rough production site. This offshore wind farm, located near the Holderness coast
of England, consists of 35 SWT-6.0-154 turbines for a covered area of 35 km square. The chosen turbine (whose identifier is *F07*) is highlighted with a blue square in Figure 13, which also provides the rest of the wind farm layout. For a north wind, the turbine performance can be numerically predicted with YALES2-BHawC, without requiring any modeling of the neighboring turbines. Operating parameters obtained numerically are compared to field data, which derive from 10-minute onsite recordings. To enhance the validation, BHawC standalone is also used to model the operating turbine. The deflections
and internal loads of the blades obtained numerically are then compared in depth.

### 5.1    Numerical setups

The geometry of the computational domain used with YALES2 and the cell-size mapping enforced within are provided in Figure 14. The mesh counts 10899287 nodes, and thus as many control volumes because YALES2 is node centered. The smallest cells are located in the turbine close vicinity and their size is set to $D/64$ to comply with the ALM common guidelines (Jha
et al., 2014). The mesh size in the wake is however set to twice this value, so that the whole mesh remains of reasonable size despite the domain extents. Close to the lower boundary, the mesh size is also set to $D/32$ to properly capture the stronger velocity gradient occurring in this region, because of the wind shear.





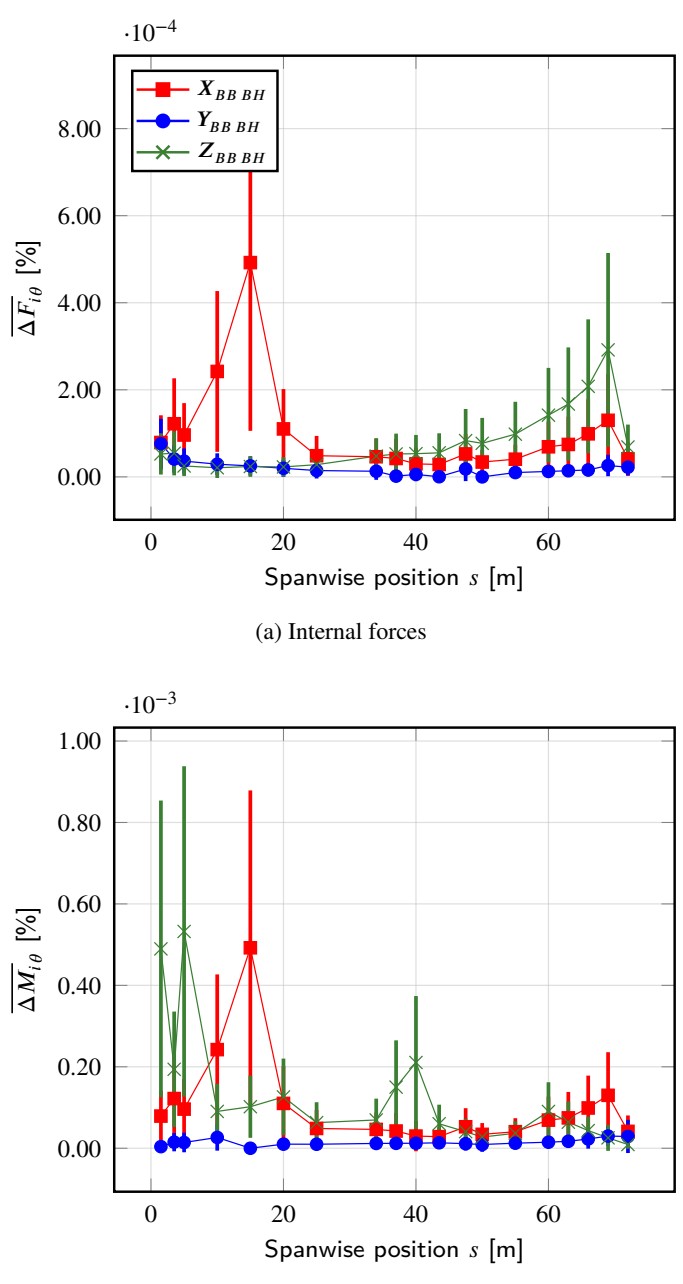

(a) Internal forces

(b) Internal moments

**Figure 12.** Mean relative error on the internal loads, given as a function of the integration interval $\Delta s$. The corresponding standard deviations are depicted as error bars.

Regarding the boundary conditions, the lateral and upper surfaces are modeled as slipping walls. On the bottom surface, a classical logarithmic wall-law including a roughness parameter is used to work around the full resolution of the atmospheric



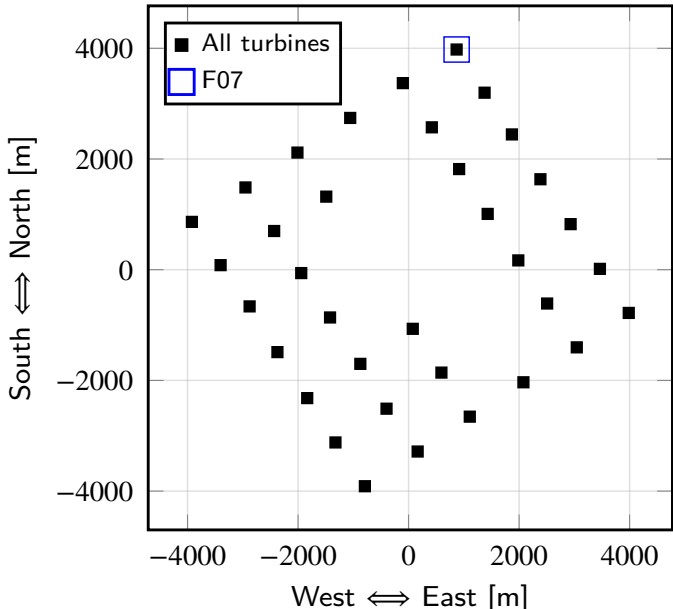

**Figure 13.** Layout of the Westermost Rough wind farm.

boundary layer. Due to the lack of reliable wind-assessment tools on the field, the inlet conditions result from a wind estimator. First, the recorded mean nacelle yaw angle is assumed to provide the wind direction. In the considered field data samples, the recorded mean yaw angles were indicating a wind coming almost exactly from the north, with all deviations being below 1 degree. Still, it should be noted that the standard deviations of the yaw angles were not zero, to reach 3.14 degrees at most. This was deemed small enough to overlook the neighbouring turbines influence on the results. Second, the wind speed $U_h$ and

turbulence intensity at hub height are computed based on the power output and pitch angles of the turbine. In this study we consider three operating points, as given in Table 1. The inlet boundary condition is finally modeled as follows:

$$u(x,y,z,t) = u'(x,y,z,t),$$
$$v(x,y,z,t) = v'(x,y,z,t),$$
$$w(x,y,z,t) = W(y) + w'(x,y,z,t), \tag{17}$$

where $(u',v',w')$ denotes a synthetic turbulent velocity-field based on the Mann model (Mann, 1994, 1998), and $W(y)$ is the

power law given by:

$$W(y) = U_h \left( \frac{y}{h_{hub}} \right)^\alpha. \tag{18}$$

The exponent $\alpha$ is set to $0.13$, as this is a typical value for offshore applications (Burton et al., 2011). The velocity fluctuations are computed prior to the actual simulations, using the turbulence simulator from International Electrotechnical Commission (IEC) (DTU Wind Energy, 2018), and then saved into periodic 3D boxes. Although anisotropic, this synthetic turbulence





remains homogeneous, meaning for instance that the turbulence damping induced by the sea in the vertical direction is not
factored in. The velocity field considered by BHawC in the BEM method is modeled identically.

**Table 1.** Mean wind speeds and target turbulence intensities at hub height on the inlet boundary.

| Wind speed | $U_h$ [m/s] | TI [%] |
|---|---|---|
| Low | 7.90 | 6.64 |
| Medium | 12.55 | 6.22 |
| High | 15.93 | 6.26 |

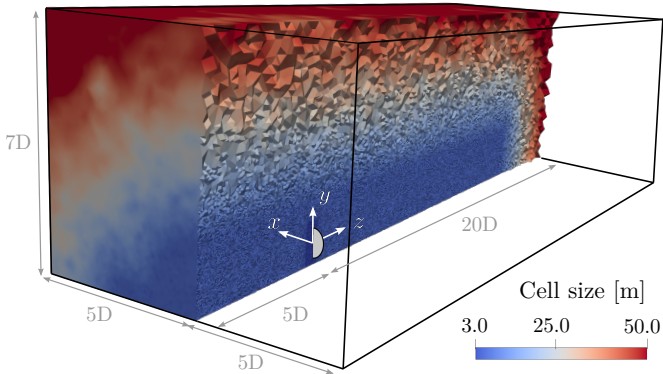

**Figure 14.** Mesh and geometry of the computational domain used to investigate the turbine F07 with YALES2-BHawC.

As additional noteworthy parameters, the air density and kinematic viscosity are respectively set to $\rho = 1.235\,\mathrm{kg.m^{-3}}$ and
$\nu = 1.448 \cdot 10^{-5}\,\mathrm{m^2.s^{-1}}$. For simplicity's sake, we also enforce $\Delta t_{Y2} = \Delta t_{BH} = 0.02$ s in all simulations, thus putting aside
the sub-stepping capability described in Section 3. For all considered wind speeds, this time-step size complies with the CFL
conditions based on the flow velocity and the particles velocity[2].

Concerning the turbine modeling, 75 aerodynamic nodes are evenly distributed along each actuator line. All the correspond-
ing isotropic mollification kernels are defined with a radius $\varepsilon = D/32$ (Jha et al., 2014). From the structural point of view,
the blades are discretized with 19 beam elements of various lengths in order to achieve a suitable representation of the elastic
axis geometry at a reasonable cost. The nacelle and tower are not represented on YALES2 side. Accordingly, the loads acting
on them are forced to zero on BHawC side, at all time. Regarding the wind turbine actual controller, it is emulated thanks to
external libraries compiled beforehand.

For completeness, it is also worth mentioning that all the structural properties of the blades, that were altered for the purpose
of verification (cf. Section 4), are now back to normal.

---

[2]A particle defined on an actuator line element should not cross more than one cell of the domain grid over a time step.





## 5.2 Results

For confidentiality reasons, all the quantitative results shown in the current section are made non-dimensional. Also, we first define the following operators:

$$\overline{\bullet}_\theta = \frac{1}{\theta_N} \int_0^{\theta_N} \bullet \, d\theta, \tag{19}$$

$$\overline{\bullet}_T = \frac{1}{T} \int_0^T \bullet \, dt, \tag{20}$$

where $\bullet$ is a quantity of interest, $\theta_N$ is defined as in Section 4.1, and $T = 600$ s is a physical duration. These operators refer
to azimuth-averaged and time-averaged results, respectively. We also introduce an additional discrete operator, referring to a mean value obtained at a given azimuth $\Theta$:

$$\overline{\bullet}_\Theta = \frac{1}{N} \sum_1^N \bullet \, (\theta = \Theta) \tag{21}$$

For each considered wind speed, Figure 15 starts by comparing the field data with the numerical results obtained with YALES2-BHawC and BHawC standalone, in terms of time-averaged rotation speed, pitch angle (first blade), and electrical
power output. Overall, average values and standard deviations are well predicted with YALES2-BHawC for all the wind speeds of interest. The time-averaged values provided by BHawC appear to be almost the same as those given by the coupled code. However, one can observe meaningful differences in the standard deviations. Those obtained with BHawC standalone are up to three times lower than those given by the coupled code. This is especially visible for the rotation speed at low wind speed.

The edgewise and flapwise bending moments at a spanwise position close to the blade root are compared for the first blade
in Figure 16. Field data were also available for such quantities thanks to strain gauges. Again, the results numerically obtained with both codes are fairly accurate. Discrepancies, reaching up to $20\%$ of the field data, are visible but the trends are very well captured.

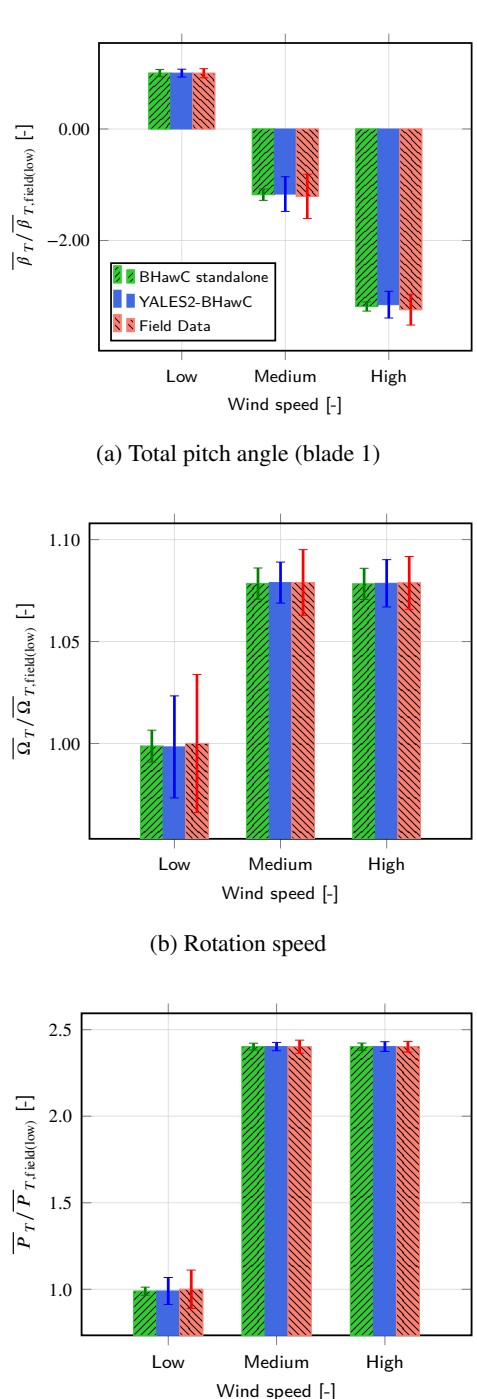

(a) Total pitch angle (blade 1)

(b) Rotation speed

(c) Power output

**Figure 15.** Comparison of the time-averaged rotation speed, total pitch angle (first blade) and power, for the three wind speeds of interest. The corresponding standard deviations are depicted as error bars.



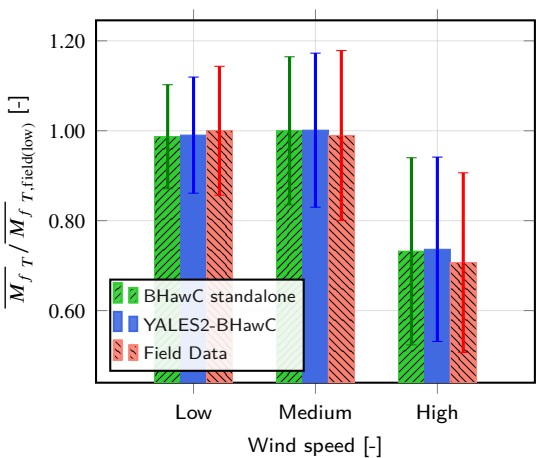

(a) Flapwise moment

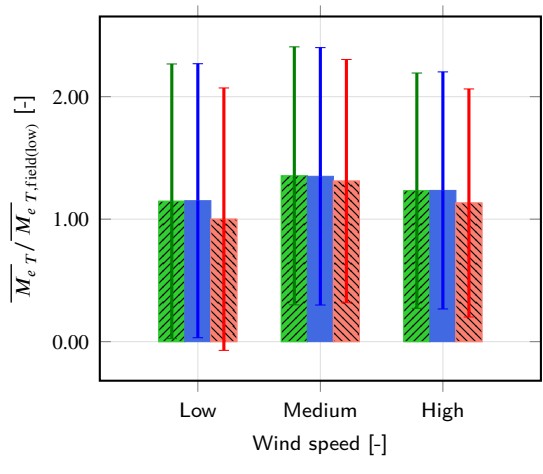

(b) Edgewise moment

**Figure 16.** Comparison of the time-averaged edgewise and flapwise bending moments, on the first blade and at the strain gauge location. The standard deviations are depicted as error bars.





In what follows, the numerical results are now compared in greater details. Despite the lack of field data for the quantities considered, the results from BHawC stand as a reference. Indeed this tool is a keystone in the design cycles of actual turbines,

and thus has been extensively validated against field data in those specific conditions (no yaw error and no upstream wake).

The first-blade tip deflection $\delta_f$ in the flapwise direction $\boldsymbol{Y}_{BB\,BH}$ is compared for all wind speeds in Figure 17. Results of both codes are depicted for $N = 10$ rotor full revolutions. The periodicity due to the rotor revolution is clearly visible in the two signals, while the turbulent inflow causes additional fluctuations. One can note significant differences between the curves. This is expected as the rotor azimuth at a given time has no reason to be the same in the two simulations. Indeed, inductions at

the aerodynamic nodes location are not computed in the same way, which leads to different aerodynamic loads and in the end to a different feedback from the controller. Besides, the turbulence seen by the rotor is not exactly the same in both codes, even initially. The reason is further developed later in this section. Yet, the amplitude of the deflections compares very well for all wind speeds.

In Figure 18, the comparison is continued in polar coordinates to visualize the distribution of discrepancies according to the

azimuthal position. To this end, we compare the mean deflection obtained at 36 azimuthal positions, based on the data from the last $N = 50$ revolutions of the rotor. Overall, the results from both codes agree fairly well at all wind speeds, with discrepancies mostly lying in the range $[-5\%, 5\%]$. No clear trend appears, meaning the coupled code does not tend to either over-predict or under-predict the results from BHawC. The local gap between the results peaks in the azimuthal range $[180°, 270°]$, for the high wind speed case, where the deflection predicted by YALES2-BHawC is on average $10\%$ higher than the one computed by

BHawC. For the medium wind speed, we can notice that the discrepancies also reach their maximum (in algebraic value) in the same azimuthal range. However, the same cannot be said for the low wind speed case, where the relative difference in the results remains constant over all the azimuth positions. This behavior is not fully understood yet. It may be related to the turbulence injected in YALES2 not being in equilibrium with the mean flow, especially close to the ground. An artificial boundary layer developing on the bottom wall from the inlet was indeed observed. Its thickness increases while moving downstream, all the

more at low wind speed to encompass the whole rotor. In the high wind speed case on the other hand, this artificial boundary layer overlaps only partially the rotor area.

To deepen the analysis, Figure 19 compares the frequency decomposition of the flapwise deflection just discussed for all the investigated wind speeds. To get representative results, the data from the last $N = 50$ revolutions of the rotor were considered as input. For each wind speed, the main peak matches the frequency $f_{rot}$, derived from the time-averaged rotation speed of the

rotor. This peak, as well as its two first harmonics, are well captured by both codes. In the three wind speed scenarios, a good overlap between the results from BHawC and YALES2-BHawC is obtained overall. Nonetheless, the peaks are slightly more diffused in the results of the coupled code. This is consistent with the standard deviations on the rotation speed being higher in YALES2-BHawC than in BHawC (cf. Figure 15b).

The time-averaged internal force $F$ and moment $M$, respectively obtained in the flapwise direction $\boldsymbol{Y}_{BB\,BH}$ and edgewise

direction $\boldsymbol{X}_{BB\,BH}$, are now compared at several spanwise positions across the first blade. The results are provided in Figure 20 for all the wind speeds of interest. The discrepancies between the results from YALES2-BHawC and BHawC are given relatively to 3 times the standard deviation in time $\sigma_T$ of BHawC results. Most of the discrepancies appear to lie within the range

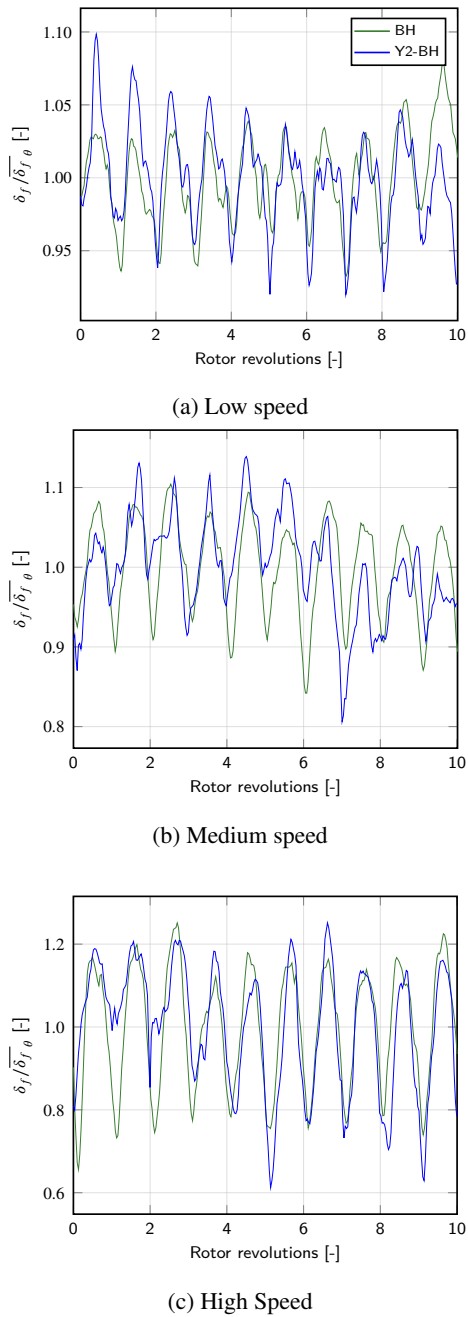

**Figure 17.** Comparison of the first-blade tip deflection $\delta_f$, in the flapwise direction, over the 10 last revolutions of the rotor, as obtained by BHawC and YALES2-BHawC.

[-10%,10%]. Still, one can observe higher gaps in the low wind speed case, especially close to the blade tip, where the loads computed by YALES2-BHawC show to be significantly lower than those coming from BHawC standalone. Nevertheless, it



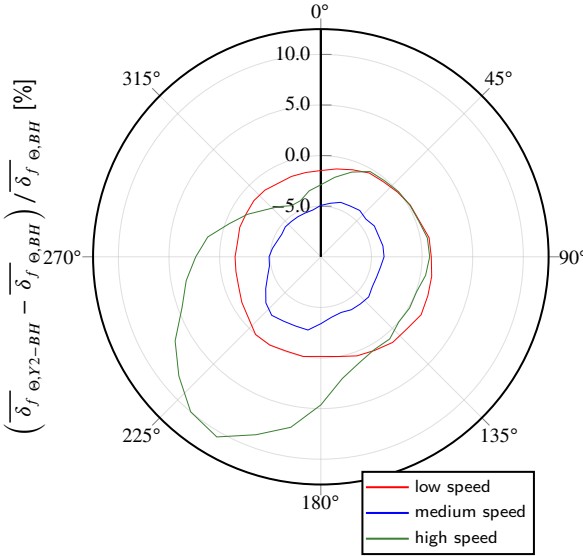

**Figure 18.** Comparison of the first-blade tip deflection $\delta_f$ in the flapwise direction, averaged per azimuthal position. The data from the last 50 revolutions of the rotor are considered.

seems important to stress that the standard deviations in time of the depicted loads, as their time average values, become smaller and smaller when heading towards the blade tip, to be almost zero at the tip.

The variability of the internal loads in time is also of great interest for engineers as it conditions the structural fatigue. Thus, the damage equivalent loads (DELs) were computed from the internal loads $F$ and $M$ just discussed. The Palmgren-Miner's rule (Miner, 1945) allows the assessment of the damage $d$ of a blade element, resulting from an unsteady internal force $F$ (or moment $M$) acting during the period $T$:

$$d = \sum_{j=1}^{N_b} \frac{n_j}{\hat{n}_j} = \frac{n_{eq}}{\hat{n}_{eq}}. \tag{22}$$

The whole variation range of $F$ is split into $N_b$ bins, which define as many discrete amplitudes $F_j$ for the included loading cycles. A *rainflow* algorithm (Matsuishi and Endo, 1968) is used to count the number of cycles $n_j$ associated with each amplitudes $F_j$. The quantity $\hat{n}_j$ denotes the maximum number of loading cycles of amplitude $F_j$ that the blade element can withstand before breaking by fatigue. Simply put, the previous law states that a pure sinusoidal load $F'$, of amplitude $F_{eq}$ and frequency $f_{eq} = \hat{n}_{eq}/T$, will lead to the same damage as the real load $F$ would. The number of cycles $\hat{n}_j$ is usually provided through a Wöhler curve (also known as S-N curve), which can be fitted by the following law:

$$\hat{n}_j = k F_j{}^m, \tag{23}$$

where $k$ and $m$ are material-dependent properties. For a wind turbine blade, even though it may vary with the deformation direction, the value $m = 10$ is traditionally used (Sutherland, 1999; Lee et al., 2012), as it refers to the glass fiber, which comes

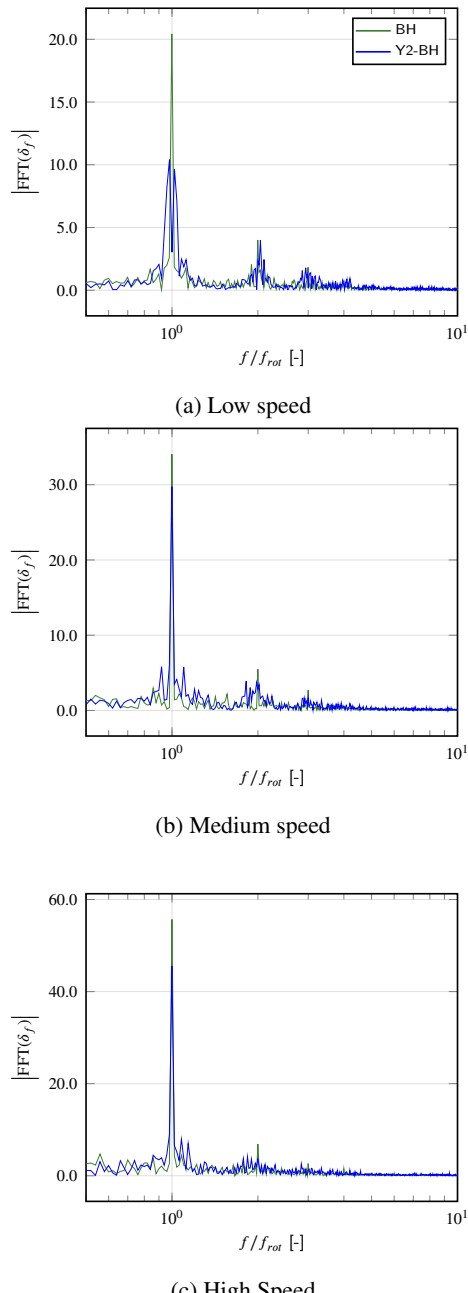

(a) Low speed

(b) Medium speed

(c) High Speed

**Figure 19.** Fast Fourier transform of the first-blade tip deflection $\delta_f$ in the flapwise direction, considering the results from the last 50 revolutions of the rotor for both YALES2-BHawC and BHawC.

in the composition of the blade reinforced parts (Mishnaevsky et al., 2017). The combination of eqs. (22) and (23) leads to the

WIND
ENERGY
SCIENCE
DISCUSSIONS

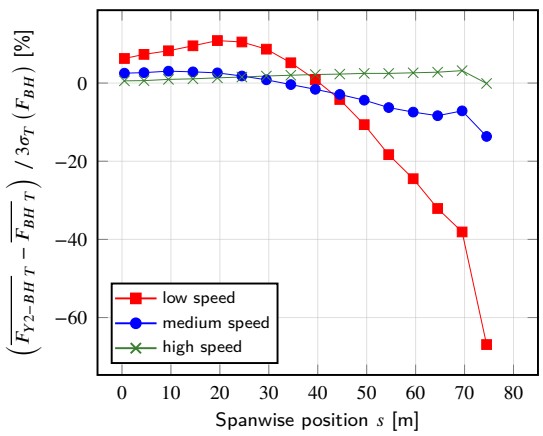

(a) Force (direction $Y_{BB\,BH}$)

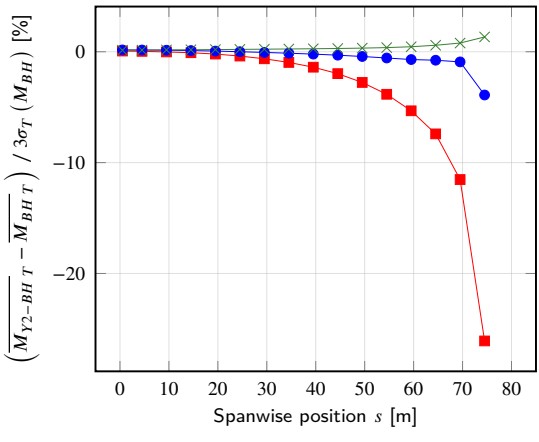

(b) Moment (direction $X_{BB\,BH}$)

**Figure 20.** Comparison of the internal loads (forces and moments) at several spanwise positions across the first blade. The quantity $\sigma_T(\bullet)$ refers to the standard deviation of the argument over the time range $T$.

following expression for $F_{eq}$ :

$$F_{eq} = \left( \sum_{j=1}^{N_b} \frac{n_j F_j{}^m}{f_{eq}T} \right)^{1/m}. \tag{24}$$

Only the choice of the frequency $f_{eq}$ remains. In order to compare the results from all cases, we set $f_{eq} = 1$ Hz, which is
475 decorrelated from the wind conditions.

The relative differences in the results obtained are reported in Figure 21. They show to be significantly higher than those observed in the results commented previously. This is particularly true for the low and medium wind speed cases, for which

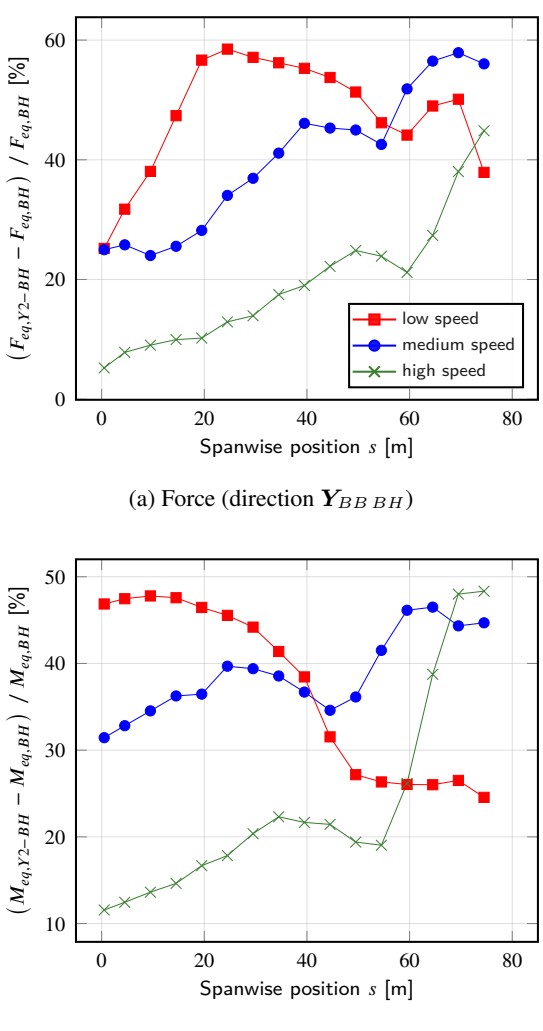

(a) Force (direction $\boldsymbol{Y}_{BB\,BH}$)

(b) Moment (direction $\boldsymbol{X}_{BB\,BH}$)

**Figure 21.** Comparison of the damage equivalents loads, computed from the internal loads (forces and moments) at several spanwise positions across the first blade.

the equivalent loads predicted by YALES2 exceed those of BHawC by about 30% on average, considering the entire blade

480 span. Two trends are also noticeable depending on the wind speed. In the medium and high wind speed cases, the discrepancies increase from the root to the mid-span region, then tend to decrease slightly before significantly increasing back when heading towards the blade tip. On the contrary, the discrepancies obtained in the low wind speed case evolve differently, especially for the flapwise equivalent moment, for which they are maximum in the blade root region.



## 5.3 Discussion on the reported discrepancies

Given their amplitude, the gaps observed in the numerical results reported in the previous sub-section likely relate to three main
factors. First, as mentioned at the end of Section 3, BHawC includes the model of Beddoes Leishman (Leishman and Beddoes, 1989) to take into account the dynamic stall effect and correct the aerodynamic coefficients accordingly. As suggested by Øye (Øye, 1991), the dynamic stall effect can be roughly translated in a time-relaxation of the aerodynamic loads. Disabling this model in BHawC was impossible in practice, as this led to the failure of the structural solver. Second, BHawC computes annular inductions, which could also weaken the shear-induced variability of the aerodynamic loads inferred afterward. Third, the turbine in BHawC does not exactly face the same turbulent structures as in the coupled code, mainly because the relative position between the turbine and the turbulence box differ. Besides, the turbulence is static in BHawC, meaning the velocity fluctuations factored in the inductions computation are the ones present in the Mann boxes. On the other hand in YALES2-BHawC, the Mann turbulence will evolve while being convected to the rotor position, especially close to the ground. As mentioned before, the Mann turbulence indeed does not include the turbulence damping in the vertical direction, which is induced by the ground. As a consequence, the turbulence intensity in the streamwise direction can also evolve between the inlet plane and the rotor plane, leading to deviations from the target values given in Table 1. In all simulations performed with the coupled code, the mean turbulence intensity in the rotor-swept area was assessed: obtained values are gathered in Table 2. It can be noticed that the value for the low wind speed case is almost twice the target value. While this last problem cannot be fully worked around, one way to address it would be to replace the Mann boxes in BHawC with velocity fields extracted from the YALES2-BHawC simulation, say one diameter upstream of the turbine (Houtin-Mongrolle, 2022).

**Table 2.** Mean effective streamwise turbulence intensities in the rotor-swept area.

| Wind speed | TI [%] |
| --- | --- |
| Low | 11.90 |
| Medium | 7.63 |
| High | 6.13 |

Considering all the aforementioned sources of discrepancies, the results given by the code YALES2-BHawC for the structural part are therefore deemed encouraging and promising. As for the prediction of the main operating parameters and performance of the turbine, the coupled code already provides very accurate results.

## 5.4 Surrounding flow

Unlike BHawC standalone, YALES2-BHawC provides insights on the flow surrounding the turbine. For all the wind speeds considered, Figure 22 presents the streamwise instantaneous velocity-field and the corresponding time-averaged field in vertical slices. All the fields shown are divided by the related target velocity at hub height $U_h$ to allow easy comparisons. One can observe the wake regions to be very different. The velocity deficit behind the turbine is indeed stronger in the low wind speed scenario, which is indicative of a significantly higher induction. This is consistent with the low wind speed being slightly





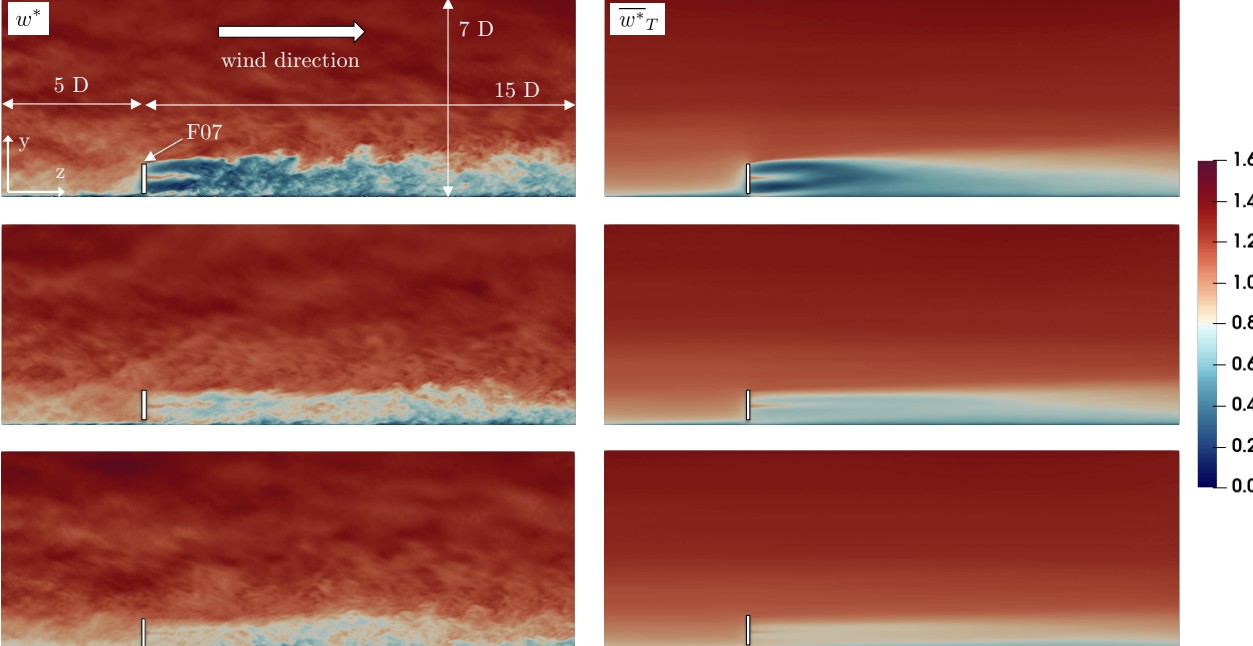

**Figure 22.** Vertical slices, including the rotor center, of the instantaneous (left) and time-averaged (right) streamwise velocity fields $w^* = w/U_h$ for the low (top), medium (middle), and high (bottom) wind speeds. Extents of the region focused on are given in a number of rotor diameter $D$.

below the rated one. Conversely, the velocity deficit obtained at high wind speed is about $20\%$. This is caused by the rotation speed being only $8\%$ higher than in the low wind speed case (cf. Figure 15b), while the wind speed is close to be $100\%$ higher. Therefore the resulting blockage effect is lower. Additionally, the cell size mapping used in the mesh shows to allow the representation of a wide range of velocity fluctuations, both upstream of the turbine and in the wake.

Figure 23 shows the same fields, but in a horizontal plane taken at hub height. The comments made previously stand but one can further notice some meandering in the wake, especially at low wind speed.

### 5.5 Computing performance

Finally, we briefly report the computing performance obtained with BHawC standalone and YALES2-BHawC. All the numerical simulations previously commented on were run on the AMD Rome partition of the Joliot-Curie supercomputer from TGCC. The simulations carried out with BHawC standalone (BH) were handled with only one CPU core, as the code is fully serial, while all the coupled simulations (Y2-BH) used 255 CPU cores: one for BHawC and the remaining ones for YALES2. The return time of each simulation, given per second of computed physical time, is reported in Table 3. In all cases, the exclusivity flag of the job scheduler was deliberately activated to achieve a better computing performance. Still, one should note that all the





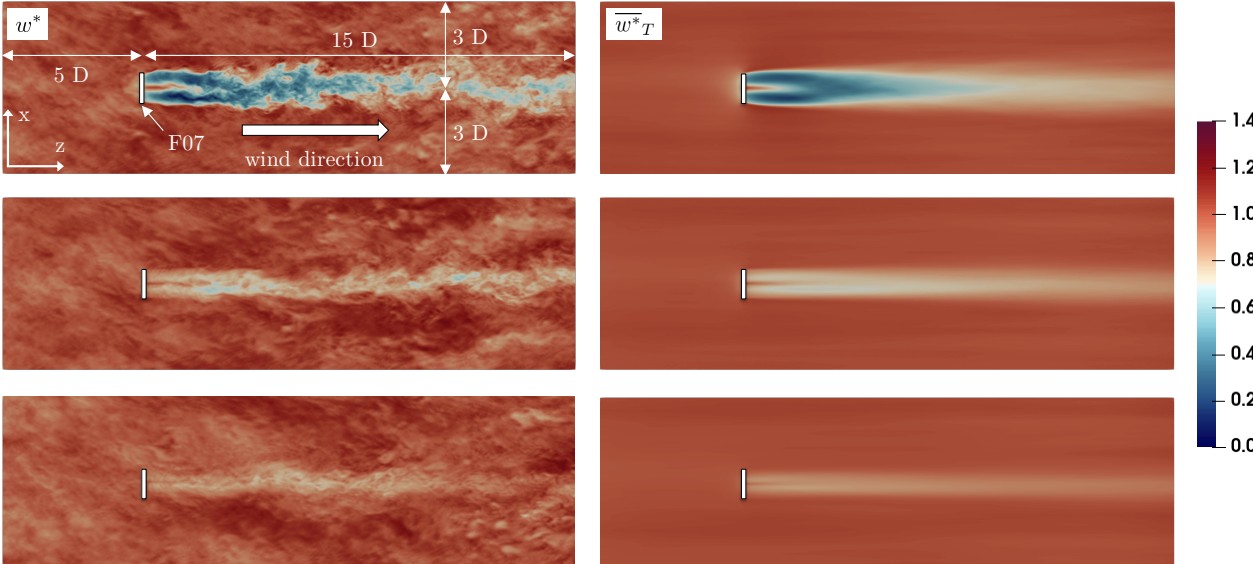

**Figure 23.** Horizontal slices, taken at hub height, of the instantaneous (left) and time-averaged (right) streamwise velocity fields $w^* = w/U_h$ for the low (top), medium (middle), and high (bottom) wind speeds. Extents of the region focused on are given in a number of rotor diameter $D$.

simulations were run only once to save CPU hours. Thus, the values given in Table 3 could be slightly biased. Yet, as expected, the return times of the coupled simulations show to be significantly higher.

**Table 3.** Return times of the numerical simulations, given per second of computed physical time.

|  | Low speed | | Medium speed | | High speed | |
|---|---|---|---|---|---|---|
|  | Y2-BH | BH | Y2-BH | BH | Y2-BH | BH |
| Return time [−] | 73.6 | 0.5 | 83.0 | 0.5 | 77.5 | 1.0 |

## 6 Conclusions

This paper first presented the details of a new aero-servo-elastic solver, obtained from the coupling between a high-fidelity massively-parallel LES solver (YALES2) and an aero-servo-elastic solver (BHawC) in use in the wind energy industry. In this coupling, YALES2 completely replace the BEM method normally used in the standalone version of BHawC, meaning that the aerodynamic loads come entirely from YALES2. Thus designed, the coupled solver allows one to get a detailed picture of the flow surrounding an actual wind turbine (or even a wind farm), where the effects of the flexible rotating blades are factored in thanks to an elastic ALM framework. Besides, the coupled solver also enable the investigation of the structural response





of each individual blade, within the limits set by a 1D beam modeling approach. Profiting from an emulation of real control strategies, the variation over time of the main operating parameters (rotor rotation speed and pitch angles) can be considered to predict a turbine power output realistically.

Then the verification of the kinematic and dynamic aspects involved in the coupled code was presented. For the former, we ensured that the position, velocity and orientation of the AL particles were computed correctly. To do so, a one-to-one comparison with YALES2 standalone was conducted. For the latter, we compared the internal loads of the first blade, computed by the coupled code and BHawC standalone. For both comparisons, the discrepancies were found to be negligible.

Finally, for the validation sake, we considered the case of an isolated turbine of the Westermost Rough offshore wind farm.
The power output obtained numerically, as well as the rotation speed and pitch angles, were compared to field data derived from 10-minute onsite recordings. Overall, an excellent agreement was pointed out. Difficulties in translating the actual incident wind into boundary conditions may explain the remaining discrepancies. To deepen the validation process, the deflections and internal loads of the blades, predicted both by YALES2-BHawC and BHawC, were compared. Again, the results agreed reasonably well, despite inherent differences in the numerical setups and in how the local inductions are computed. Especially,
the lack of a dynamic stall model in YALES2-BHawC could explain the differences in the damage equivalent loads across a blade. Still, on average the flapwise deflections at the blade tip were shown to be only $5\%$ apart, and their frequency spectra overlapped well.

Finally it seems important to stress that, from an industrial perspective, the coupled code YALES2-BHawC remains too demanding in memory and CPU resources, and too time-consuming to be used directly in actual design cycles. However, as field
data do not always provide insights on the physics involved, this tool can be used advantageously to help designing, calibrating, and certifying lower-order approaches such as the BEM method or wake engineering models. Because computationally affordable and predictive, those are indeed used on a daily basis in the industry to assist mechanical design and siting studies.

**Appendix A:  Coupling between YALES2 and BHawC: technical details.**

No specific directive was included in the YALES2 library directly. Alternatively, an additional library is called dynamically
from YALES2 to manage the communications with BHawC. This generic approach can be used to design a subsequent coupling between YALES2 and a BHawC-like code, such as OpenFAST (Jonkman et al., 2005; National Renewable Energy Laboratory, 2022) or HawC2 (Larsen and Hansen, 2007). Moreover, this new library was also coded in Fortran 90 to make all the subroutines of YALES2 readily accessible. However, this approach was not used with BHawC, which stands as an executable once compiled. For simplicity, communication directives were therefore written in BHawC directly. This architecture is outline in
Figure A1.

All the required communications between the two codes were handled by means of the Message Passing Interface (MPI) library (Message Passing Interface Forum, 2021) (see Figure A2). So defined, the coupling allows to model both an isolated wind turbine as well as a wind farm. To each turbine corresponds a single BHawC process. Conversely, in YALES2, a turbine is known by all the MPI processes involved in the partition of the computational domain. This straightforward approach was





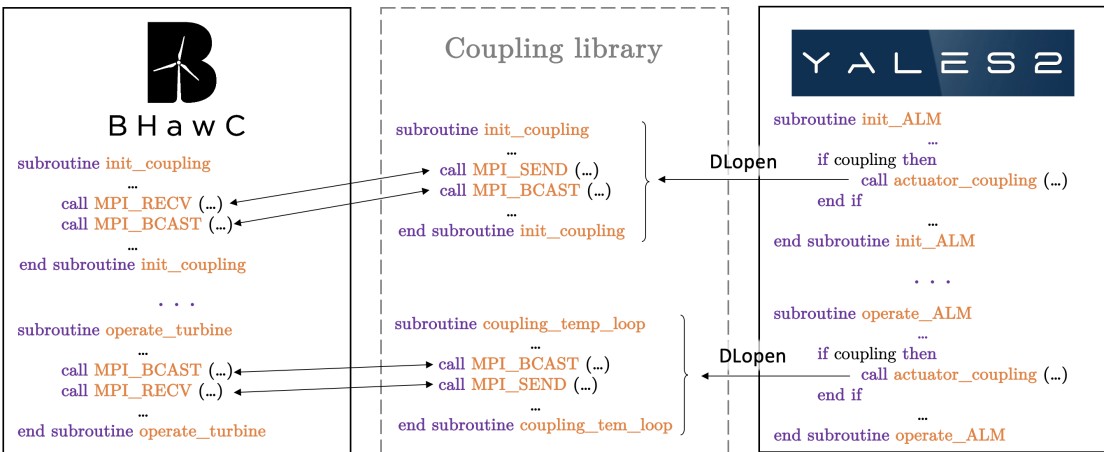

**Figure A1.** Sketch of the coupling architecture. A external library is called dynamically from YALES2 to manage all the communications with BHawC.

chosen to easily comply with the mesh adaptation capabilities of YALES2, which may change the partition and cell groups
initially known by a YALES2 process. Yet, this feature was not used in this work. An additional MPI communicator is created
for each turbine, which encompasses the corresponding BHawC process and all YALES2 processes. In each communicator,
the master rank is the BHawC process, which can thus be easily identified thereafter. All data exchanges between BHawC
and YALES2 are carried out via these communicators. Data from BHawC are broadcasted to all YALES2 processes, while

transfers from YALES2 to BHawC are achieved via point-to-point communications. At the beginning of the simulation, a
YALES2 process is chosen to perform the latter. For convenience, this process is called *sub-master* of the communicator. To
balance the workload, a YALES2 process can be the sub-master of only one turbine communicator at most. The code will
deviate from this scenario only when the turbines outnumber the YALES2 processes. Yet, this never happens during realistic
simulations: the mesh size increases with the number of turbines, thus requiring the use of additional cores. Besides, each

turbine is controlled independently from the others. No specific MPI communicator is therefore needed between the BHawC
processes.



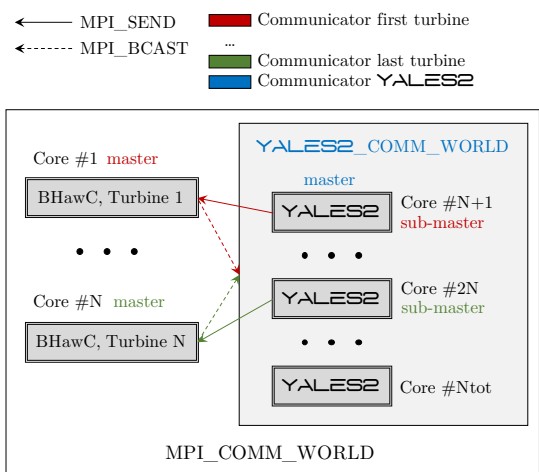

**Figure A2.** Sketch of the MPI communications involved in the coupling between YALES2 and BHawC.

*Author contributions.* E. Muller and S. Gremmo both contributed to the following aspects of this work: conceptualization, methodology, software development, formal analysis, verification, validation, simulations setup & post-processing, and writing of the original draft. Valuable support was provided by F. Houtin-Mongrolle, B. Dubob and P. Bénard regarding the conceptualization, methodology, software development,

as well as for reviewing the original draft. The field data were provided by B. Duboc. Finally, B. Duboc and P. Bénard were also in charge of the supervision, project administration, and funding acquisition.

*Competing interests.* B. Duboc declares that he was a full-time employee of Siemens Gemesa Renewable Energy at the time this work was carried out.

*Acknowledgements.* First, we would like to thank the European Union and the Normandy region (France) for funding this work as part of the

WAKE OP project. We acknowledge PRACE for awarding us access to Joliot-Curie at GENCI@CEA, France. This work was also granted access to the HPC/AI resources of TGCC under the allocation 2021-A0102A11335 made by GENCI, and to CRIANN resources under the allocation 2012006. Finally, we also want to acknowledge Ørsted A/S for sharing the Westermost Rough field data used in this work.



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
