# Peer review of "Field data based validation of an aero-servo-elastic solver for high-fidelity LES of industrial wind turbines"

_Wind Energy Science, 2023_

## Author Comment (AC1)

**Reply to reviewer 1**

Etienne Muller[1], Simone Gremmo[1], Félix Houtin-Mongrolle[1], Bastien Duboc[2], and Pierre Bénard[1]

[1]Univ Rouen Normandie, INSA Rouen Normandie, CNRS, CORIA UMR6614, 675, avenue de l'Université, Saint-Etienne-du-Rouvray, 76801, France
[2]Siemens Gamesa Renewable Energy, 685 Avenue de l'Université, Saint-Etienne-du-Rouvray, 76801, France

**Correspondence:** Etienne Muller (etienne.muller@coria.fr)

The authors would like to thank the reviewer for the thorough review and many good points made, which will improve the paper significantly. In the following, the reviewer's comments will be given in black font, while the answers from authors are given in blue.

This article presents an interesting comparison of statistical quantities between a field experiment, LES+ALM+BHawc, and BEM+BHawC. The paper is generally well written and provides many details about the setup. I enjoyed seeing results from a field campaign as part of the comparison. Some parts of the paper need clarification and a better discussion. I have the following recommendations to improve the manuscript before publication.

1. The authors are comparing two methods (BEM vs LES), but they have quite different inflows between the methods. The authors need to include plots of wind speed (inflow) as function of height and turbulence intensity as function of height for the different cases. The inflow will be the first source of difference between the methods and the authors should discuss this in more depth.

   As emphasized in sub-section 5.3 of the manuscript, there are indeed several differences in inflows respectively used for the LES and BEM simulations, and those could contribute significantly to the discrepancies observed in the results. Even though the mean velocity profile enforced at the inlet of the CFD domain is the one used in BHawC, this profile is expected to distort slightly as it is not in balance with the injected synthetic turbulence and the other boundary conditions (especially the one at the ground). The injected velocity fluctuations are also expected to be altered for the same reason. These are typical shortcomings of synthetic turbulence, especially for wall-bounded flow. To better illustrate this, figures showing the time-averaged streamwise velocity fields probed one and two diameters upstream of the turbine in YALES2-BHawC were added to the manuscript for all wind speed cases. Similar additional figures are also provided for the turbulence intensity.

2. The authors select 75 aerodynamic nodes along the blade and value of epsilon for their simulations. The values selected (eps/D) are not expected to provide an optimal representation of the blade loading from an ALM. I recommend the authors to expand this discussion and to highlight some of the limitations from the ALM at the given resolutions.

   The parameters used for the ALM framework in YALES2 were mainly selected by following existing guidelines. Regarding the cell size $\Delta_{grid}$ in the rotor region, it was uniformly set to $\Delta_{grid} = D/64 \leq R/30$, where $D$ and $R$ are the rotor

diameter and radius respectively. The spacing $\Delta_b$ between the particles on the actuator lines was set to $\Delta_b = R/75 \leq \Delta_{grid}$. Finally, the radius $\varepsilon$ of the Gaussian kernel was constantly set to $\varepsilon = 2\Delta_{grid} = D/32$ along the blade span. These combined values of $\varepsilon$ and $\Delta_b$ may lead to some accuracy reduction in the vicinity of the blade tip and root if the width $w$ of the blade elements, considered in the calculation of the aerodynamic forces, is set as $w = \varepsilon$. In our case however, we set $w = \Delta_b$, leading the particle-wise aerodynamic forces to consistently reduce in amplitude if the number of blade elements is increased. In this work, the value of $\Delta_b$ was chosen to prevent interpolation errors when getting the aerodynamic coefficients from the blade look-up table. Indeed, 75 spanwise positions were referenced in the latter, which correspond exactly to the position of the defined particles. Besides, this value of $\Delta_b$ allows also to match the number of aerodynamic nodes used in BHawC, as well as their respective positions. To reflect this discussion, the selected settings for the ALM framework were also further commented in the relevant part of the article.

**Specific comments:**

"A substantial advantage of this code is the possibility to emulate the actual controller of industrial wind turbines."

*Comment*: Have you been able to successfully use an industry controller? Industry typically does not share their controllers. It is possible to use their controllers in other codes as well, but researchers do not have access to these controllers. I would recommend that the authors remove/modify this statement as it might be misleading.

This work relied on a complete emulation of the actual controller of the investigated wind turbine (SWT-6.0-154 from Siemens Gamesa Renewable Energy). The comparison, between the numerical results and the field data, of the first-blade pitch angle response would have not been possible otherwise. Besides, BHawC cannot be run without a controller component (as discussed in former Section 4, now moved to the appendix). The external library containing the controller logic was provided by Bastien Duboc, co-author of this paper and employee of Siemens Gamesa Renewable Energy. Still, it is worth mentioning that this library was used as a black box in all simulations performed, as the authors never had access to the controller source code. The statement has been reworded to reflect this clarification.

"The latter is usually chosen as twice the maximum cell-size encountered in the rotor region."

*Comment*: Can you please expand on the description of epsilon chosen for this work? There is a large dependency of blade loading on epsilon. Please specify the actual value of epsilon (as physical length scale) and why the choice was made. Also, epsilon/dx=2 is typically not enough resolution to properly resolve the flow field around the ALM and the loading is affected by this.

The choice of $\varepsilon$ in this work was motivated by two main reasons.

First, the value $\varepsilon = 2\Delta_{grid} = D/32$, along with a constant cell size in the rotor area, is widely used in the literature, even in fairly recent research works. A lower value of epsilon, say $\varepsilon = \Delta_{grid}$, was often reported to lead to numerical instabilities in the vicinity of the blades and in the near wake, thus preventing an accurate prediction of both the local flow field and the blade loads. On the other hand, a higher value of $\varepsilon$, still used along with a constant cell size in the rotor area, was shown to lead to an over-predicted generated power.

Second, this work addresses the question of using LES simulations directly in an industrial context to further support improvements of lower-fidelity approaches, such as BEM-like methods or analytical wake models. As stated in the manuscript conclusion section, the LES approach, from an industrial perspective at least, is still considered very costly due to the required CPU resources. Therefore, some trade-offs must be found between achievable accuracy and simulation cost. It was indeed shown in the literature that one should ideally opt for a varying value of $\varepsilon$ along the blade span, to comply with the local length $c$ of the blade section chord. The value of $\varepsilon = 0.25\,c$ was reported several times as a suitable choice. Yet, as one must still comply with $\varepsilon \geq 2\Delta_{grid}$, the consistent mesh size in the rotor region would become prohibitive from an industrial point of view.

"Yet, the code can only handle one turbine at a time as it considers only one incident flow field. The simulation of an operating wind farm remains feasible by considering different incident flows for each turbine, mimicking at best the local flow properties."

*Comment*: I recommend the authors to remove this statement or elaborate. This statement applies to any code, each turbine has different inflow, so a new instance should always be used per turbine.

This statement was intended as way to emphasize that BHawC, when relying on an appropriate inflow model (such as a DWM model), can still be used to assess the performance of a wind farm for given free-stream wind conditions, as well as for predicting the loads applied on each individual turbine. Still, it was here implied that all inflows remain fully decoupled, and possibly inconsistent, thus adding additional biases in the results. Besides, setting an inflow representative of multiple wakes combination may not be straightforward, as one need, for instance, to know which wakes are expected to combine prior to the simulation. The statement has therefore been elaborated to make the latter comment fully explicit.

"We expect this issue to be at least partially removed by implementing an appropriate dynamic stall model in YALES2. Indeed, simulations carried out with the standalone version of BHawC showed a link between the divergence of the structural solver and the dynamic stall phenomenon model activation. It is emphasized that the induction is close to being maximum in the mentioned wind speed range, leading to a strong coupling between the structure and the ambient flow. Therefore, the relaxation of aerodynamic forces induced by the dynamic stall phenomenon is likely to be essential in these cases."

*Comment*: This seems to be an interesting hypothesis. However, it is not the focus of the paper. I recommend the authors to avoid these statements unless they relate to the work done in the manuscript.

The statement, which is indeed not supported by solid research yet, has been replaced to introduce the need for a more comprehensive and detailed future analysis of the issue.

Sections 4.1 and 4.2

*Comment*: These sections provide interesting checks to ensure that the code is working as expected. However, they do not add much value to the manuscript and should be moved to an appendix.

As suggested, the Section 4 of the manuscript was moved to an appendix. Some parts of the article text were slightly adjusted to comply with this change.

"In all simulations performed with the coupled code, the mean turbulence intensity in the rotor-swept area was assessed: obtained values are gathered in Table 2."

*Comment*: There is a significant mismatch between the codes. I recommend the authors to include profiles of wind speed and turbulence intensity from the different methods.

This part was reworked. The method used to assess the ambient streamwise turbulence intensity in the coupled simulations was revised. Indeed, the values reported in Table 2 were obtained by considering statistics collected in the rotor-swept area, and computed as the average of local turbulence intensities (i.e. ratios of the local streamwise RMS velocities over the *local* streamwise time-averaged velocities). This led to an unfair comparison with the target values given in Table 1, which were simply derived from the ratios of the local streamwise RMS velocities over the streamwise time-averaged velocity *at hub height*. In addition, the revised TI values now given in Table 2 were extracted one and two diameters upstream of the rotor as well, to better illustrate the variations. This was deemed more representative as the blades rotation is expected to increase the turbulence intensity.

"Considering all the aforementioned sources of discrepancies, the results given by the code YALES2-BHawC for the structural part are therefore deemed encouraging and promising. As for the prediction of the main operating parameters and performance of the turbine, the coupled code already provides very accurate results."

*Comment*: I recommend the authors to reevaluate these statements. All these discrepancies suggest that there is a substantial amount of error cancellation between models going on.

As highlighted in the reviewer's comments, inherent differences in the inflows considered in BHawC and YALES2-BHawC are expected to lead to *discrepancies* in the numerical results. Especially, one can expect extra fluctuations of the loads in YALES2-BHawC as the flow velocity retrieved at the particles location will be more turbulent (because of blade-added and ground-added turbulence). This holds particularly true for the low wind speed case as the corresponding TSR value is significantly higher. For instance, Figures 17 and 19 of the original manuscript emphasized a richer spectral content for the blade tip deflection in the case of YALES2-BHawC results. Consistently, the damage equivalent loads are higher for YALES2-BHawC, as illustrated in Figure 21. However, time-averaged loads were expected to remain very similar between YALES2-BHawC and BHawC, assuming that the power law used at YALES2 inlet would not distort too much when convected downstream. As illustrated in the figures added to the manuscript (to answer the reviewer's first main recommendation), the previous assumption is rather well met for the medium and high wind speed cases, and the differences in time-averaged internal loads reported in Figure 20 show to be mostly below 5%.

Regardless, the current work highlights the challenges of comparing high-fidelity and medium-fidelity simulation methods. In particular, it proves to be difficult to have the same inflow conditions in both approaches and this complicates the comparison. Nevertheless, the results presented so far can be seen as a first encouraging step towards a full validation of YALES2-BHawC.

Once again, the authors would like to thank the reviewer for providing very constructive comments.

---

## Author Comment (AC2)

**Reply to reviewer 2**

Etienne Muller[1], Simone Gremmo[1], Félix Houtin-Mongrolle[1], Bastien Duboc[2], and Pierre Bénard[1]

[1]Univ Rouen Normandie, INSA Rouen Normandie, CNRS, CORIA UMR6614, 675, avenue de l'Université, Saint-Etienne-du-Rouvray, 76801, France
[2]Siemens Gamesa Renewable Energy, 685 Avenue de l'Université, Saint-Etienne-du-Rouvray, 76801, France

**Correspondence:** Etienne Muller (etienne.muller@coria.fr)

The authors would like to thank the reviewer for the thorough review and many good points made, which will improve the paper significantly. In the following, the reviewer's comments will be given in black font, while the answers from authors are given in blue.

The paper presents a field data-based validation of an aero-servo-elastic solver coupled with LES simulations. The topic is very interesting and relevant for the readers of Wind Energy Science. The article is well-written. The authors provided detailed explanations regarding how to couple the LES tool YALES2 with the aeroelastic tool BHawC.

I only have some minor comments as follows:

1. page 2. Please define 'CFD'

   The definition was added.

2. page 2. '... HawC2...' should be HAWC2

   This has been corrected.

3. "To the authors' knowledge, (Gremmo et al., 2022) is the only reference where a code with these capabilities has been reported." Can the authors elaborate on what capabilities they have? As far as I know, there are also examples of coupling between EllipSys and HAWC2, which can be easily found in the literature.

   This statement was not referring to the Elastic-Actuator-Line kind of coupling, which indeed led to several publications in the litterature already. The authors were rather trying to emphasize that YALES2-BHawC can additionally embed the actual logic of a SGRE wind turbine controller, through the dynamic loading of a specific library. This part of the introduction was slightly reworded to clarify it. An additional reference has also been added in the introduction section of the manuscript, regarding an existing coupling between EllipSys 3D and HAWC2, which was indeed missing.

4. page 8. "First, it cannot work without the structural solver also having at least the blade element theory implemented within. In other words, a standalone structural solver is not suited for this strategy." Can the authors elaborate more? Why wouldn't the standalone structural solver without the blade element work?

   The use of a coupling approach, where the flow velocities are fed back to the structural solver, implies that the aerodynamic loads need to be calculated internally using these velocities. As a consequence, the structural solver needs to

25      integrate a BEM module to convert the velocity into an aerodynamic force, and so "... a standalone structural solver is not suitable for this strategy". The sentence has been edited to improve readability

5. Figure 8. I don't really understand this figure. It seems the mean error is so small? Is it true?

The error is indeed expected to be so small. Here two configurations are compared and the difference lies in the structural model:

30      (a) The YALES2 configuration has a fully rigid structure, which is infinitely rigid.

(b) YALES2-BHawC has a very high structural stiffness (cannot be set to infinity for numerical reasons).

The point of this graph is to emphasise that, despite this difference, a meaningful comparison can be made because the residual flexibility of the structure is very small, resulting in a very small difference (i.e. nondimensional mean error) in particle position between the two configurations. The caption of Figure 8 is modified to explicitly mention the two
35      configurations.

Once again, the authors would like to thank the reviewer for providing very constructive comments.